



**Geomorphic analysis and fluvial incision rates from valley-filling lava flows:**
**implications for the Quaternary morphotectonic evolution in the Moroccan Massif**
**Central and Middle Atlas**
Ahmed Yaaqoub[1], Abderrahim Essaifi[1], Romano Clementucci[2,3], Paolo Ballato [2], Rachid Zayane [1], Claudio
Faccenna[2], Carolina Pagli[4]
[1]Département de Géologie, FSSM, B.P. 2390, Université Cadi Ayyad, Marrakech, Maroc.
[2]Dipartimento di Scienze, Università degli studi Roma Tre, Roma, Italia.
[3]Department of Earth Sciences, ETH Zurich, Zurich, Switzerland
[4]Dipartimento di Scienze della Terra, Università di Pisa, Pisa, Italia.
*Correspondence to: Ahmed Yaaqoub (*ahmed.yaaqoub@ced.uca.ma)
## Abstract
Fluvial dynamics is one of the main surface processes that shape the Earth's topography.
Geomorphic records, such as fluvial terraces, play a crucial role in reconstructing the history of landscapes
and deciphering the complex interactions among tectonic activity, lithology, and surface processes, which
are primarily controlled by climate. This is also valid in valleys characterized by the emplacement of effusive
volcanic rocks that are generally more resistant to erosion, and hence have a high preservation potential,
and are easier to date than alluvial deposits. Valley-filling volcanic rocks, thus, represent ideal geomorphic



markers to estimate the magnitude and the spatio-temporal pattern of fluvial incision and associated
forcing mechanisms.

24        In this study, we combine fluvial incision rates on dated lava flows emplaced in the valleys of the

Moroccan Massif Central and Middle Atlas with DEM-based geomorphic analysis to gain insights into the
Quaternary landscape evolution. The results show that incision rates are in the order of 0.01 and 0.1 mm yr$^{-}$
$^{1}$ for the Middle Atlas and the Massif Central respectively. This spatial discrepancy in incision rates agrees
with geomorphic metrics, with lower rates within the low topographic relief landscape and higher rates (up
to one order of magnitude) along its margins that are highly dissected by fluvial incision. The comparison
between our data and published incision rates in the northeastern flank of the Middle Atlas suggests that
the eastern flank of the Middle Atlas accommodates active tectonic shortening. Furthermore, our analysis
indicates that lithology and climate may not be the primary factors controlling the observed spatial
variation in incision rates between the Middle Atlas and the Massif Central. Instead, surface uplift, which is
probably related to forebulge flexural uplift enhanced by dynamic mantle-related uplift, could have
triggered relatively high incision rates in the Massif Central. Ultimately, we conclude that a significant
proportion of the topographic relief in our study area has been generated before the lava emplacement
(i.e., earlier than 2.85 Ma).
**Keywords:** Incision rates, lava flows, Middle Atlas, Massif Central, Morphotectonic evolution, Quaternary
1. Introduction

The evolution of the Earth´s landscapes is the product of the perpetual interactions between

tectonics, which tend to construct topography through rock uplift, and surface processes, which work to
reshape and lower it (e.g., Ahnert, 1970; Kirby and Whipple, 2012; Whipple and Tucker, 1999; Whittaker,



2012; Wobus et al., 2006). Fluvial incision is one of the most important geological exogenic process that
drives the evolution of landscapes. Therefore, quantifying fluvial incision rates is pivotal in understanding
the pace and patterns of landscape development (Wolff et al., 2018). Bedrock rivers are sensitive to
changes in boundary conditions such as uplift, rock erodibility, base-level elevation, and climate, and
bedrock rivers communicate base-level changes to the whole catchment, dictating the pattern of erosion
and deposition (Boulton, 2020; Lanari et al., 2020; Whipple et al., 2013). As such, fluvial landforms
constitute archives that encode valuable information about tectonic and climatic conditions of a given
region (Demoulin et al., 2017; Evenstar et al., 2020). With the advent of Quaternary geochronology, fluvial
terraces are increasingly used to quantify incision and uplift rates and to infer tectonic and climatic histories
(Agharroud et al., 2021; Burbank and Anderson, 2011; Fuller et al., 2009; Lavé and Avouac, 2001; Pastor et
al., 2015; Pazzaglia, 2013). Similarly, many volcanic terrains provide good location to measure incision and
denudation rates (Clementucci et al., 2022; Ott et al., 2018). Fluid lava flows emplaced along the preexisting
valleys create an isochronous surface. The post-emplacement undercutting by fluvial incision and the
formation of fluvial strath terraces provides the possibility to quantify rates of erosion/incision. Many
studies took advantage of this interaction river-lava to quantify fluvial incision rates and constrain
landscape evolution (e.g., (Allen et al., 2011; Bridgland and Westaway, 2014; Demir et al., 2007; Ott et al.,
2018; Schildgen et al., 2007).
The Middle Atlas Mountains is the northeast branch of the Atlas system of Morocco. It is a Cenozoic
intracontinental belt that resulted from the inversion of an aborted Triassic-Jurassic rift that affected the
Hercynian terrains of the Moroccan Meseta (Charrière, 1990; Mattauer et al., 1977; Michard, 1976). The
polyphased tectonogenesis of the Middle Atlas, is related to the Africa-Europe convergence, and took
place in the Neogene before the late Miocene and during the Pliocene (Charrière, 1984, 1990). Tectonic
shortening, however, is too moderate to solely account for the high elevation of the belt (Arboleya et al.,
2004; Gomez et al., 1998). A long-wavelength mantle-driven uplift is thought to have contributed to the
generation of high topography as also suggested for other mountain belts of Morocco (e.g., Babault et al.,
2008a; Clementucci et al., 2023a, b; Frizon de Lamotte et al., 2009; Lanari et al., 2023b; Miller and Becker,

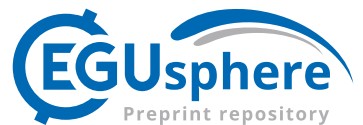

2014; Missenard et al., 2006). This interpretation is supported by the occurrence of a thinned mantle
lithosphere associated with scattered alkaline volcanism that stretches from the Siroua Massif in the Anti-
Atlas towards the Mediterranean sea (Teixell et al., 2005). Particularly, in the Middle Atlas and the Massif
Central, Quaternary lavas were first emplaced along river valleys for tens of kilometres, and then incised by
re-established rivers to form gorges. The uplift of Messinain (~7 Ma) shallow marine deposits in the Middle
Atlas serves as a significant marker for constraining the lower limit of the timing of dynamic uplift (Babault
et al., 2008). However, the upper limit of this timing is debated, and it is not clear if mantle-related uplift
keeps sustaining the high topography of the Atlas system throughout the Quaternary (Lanari et al., 2022).
In this study, we took advantage of the distinctive geomorphic interaction between dated lava
flows and main rivers in the Middle Atlas and the Moroccan Massif Central to calculate Quaternary fluvial
incision rates. We combined the estimated post-eruption fluvial incision rates with topographic and fluvial
DEM-based analysis and pre-existing data (e.g., [10]Be-dervied denudation rates, Clementucci et al., 2023b) to
discuss the main factors that control fluvial incision and to investigate the tectonic and geomorphic
evolution of this area of Morocco during the Quaternary.

## 2. Regional setting

The Atlas system is an intracontinental mountain belt that developed in the African plate of the
Cenozoic Alpine belt (Fekkak et al., 2018; Gomez, 1996; Lanari et al., 2020). Geographically it spans from
the Atlantic margin of Morocco to the Mediterranean coast of Tunisia for over 2000 km and forms a major
morphologic barrier between the Sahara domain and the western Mediterranean Sea (Frizon De Lamotte et
al., 2000). The High Atlas and the Middle Atlas represent the Moroccan part of this orogenic domain, which
is bordered to the north by the alpine-type Rif belt and to the south, by the Paleozoic Anti-Atlas range (Fig.
1). The Mesetan domains (Western and Eastern Meseta) are made of Paleozoic terrains and their Mezo-
Cenozoic cover (Hoepffner et al., 2005; Michard et al., 2023a) (Fig. 1 and 2). The Western Meseta consists



of several Paleozoic massifs deformed during the Variscan orogeny, separated by Meso-Cenozoic
sedimentary series such as the Cretaceous-Cenozoic Phosphate plateau (Essaifi et al., 2014; Michard et al.,
2023b; Ouanaimi et al., 2019). The Moroccan Massif Central, which is the largest massif of the Western
Meseta, exhibits high elevation despite the lack of evidence of Cenozoic tectonic deformation (Barbero et
al., 2011; Clementucci et al., 2023b; Yaaqoub and Essaifi, 2023).

The Middle Atlas is the NE trending branch of the Moroccan Atlas system. It runs obliquely from the

WSW-ENE striking High Atlas near Beni Mellal in the Southwest until Taza in the Northeast where it is
covered by the Neogene-Quaternary Taza-Guercif basin (Fig. 2). The post-Paleozoic geodynamic evolution
of the Middle Atlas started with the Triassic and Jurassic rifting associated with the opening of the Atlantic
and Tethys oceans (Beauchamp et al., 1996; Frizon De Lamotte et al., 2008; Jacobshagen, 1988). The
extensional kinematic was controlled by zones of weakness inherited from the Variscan orogeny that
affected the Meseta basement (Mattauer et al., 1977). Tectonic inversion in the Middle Atlas Mountains,
driven by the convergence of the Africa and Eurasia plate, began in the late Cretaceous, with the main
phase of uplift occurring during the Neogene (Charriere, 1984; Gomez et al., 2000; Lanari et al., 2023b).

Geomorphologically, the Middle Atlas can be divided, from the NW to the SE into two distinctive

domains (Martin, 1981), the Tabular Middle Atlas (TMA) and the Folded Middle Atlas (FMA) (Fig. 1 & 2). The
TMA is a weakly deformed area that consists mainly of Triassic and Liassic rocks. This sector exhibits a flat
and high topography (mean elevation ~1500m) and sub-horizontal strata, which lie unconformably over the
Palaeozoic basement of the Western Meseta, and it is traversed by a NE trending sinistral strike-slip fault,
the Tizi-N-Tigheten fault. The FMA presents a high relief topography with four major narrow fault-related
anticlinal ridges separated by three broad synclines (Colo, 1961; Gomez et al., 1998; Zafaty et al., 2023; Fig.
2). In addition to Triassic and Liassic sequences, thick and deep basin sequences related to tectonic
subsidence were deposited in the FMA during the middle Jurassic (Charrière et al., 1994; Du Dresnay,
1988). Two major faults, namely the North Middle Atlas Fault (NMAF) that constitutes a limit with the TMA,
and the South Middle Atlas Fault (SMAF), which separates the Middle Atlas from the peripheral foreland



(Missour basin), border the FMA. These faults are considered as the Middle Atlas paleo rift-bounding faults
(Fedan, 1988). Accordingly, the FMA represents the paleo-rift basin whereas the TMA and Missour/High
plateaux are its NW and SE margins respectively (e.g. Beauchamp et al., 1996).

The sedimentary record of the Middle Atlas is made of three megacycles separated by two

sedimentary gaps (Boumir et al., 2023; Charrière, 1990; Frizon De Lamotte et al., 2008; Zafaty et al., 2023).
The first megacycle (Trias-Malm) starts with Triassic continental deposits associated with tholeiitic basalts
(Fig. 3), then the sedimentation becomes dominantly marine in the Lias-Malm with shallow-water
carbonates accumulated in continental shelves and deep-water carbonates being deposited in subsiding
basins (El Arabi et al., 2001). The second megacycle (Barremian-Eocene) corresponds mainly to littoral
deposits with frequent evaporitic episodes in local depocenters along the western part of the Middle Atlas
(Fig. 2). Finally, the last cycle (Tortonian to Plio-Quaternary) occurs mostly along the northern slope of the
range and includes fluvio-alluvial continental deposits and shallow to moderately deep marine deposits
(Fig. 2 and 3).

Restored geological cross-sections in the Middle Atlas yield a limited amount of Alpine crustal

tectonic shortening of about 5km (Arboleya et al., 2004; Gomez et al., 1998). This moderate value is
consistent with the low crustal root thickness of ~30 km estimated by geophysical studies (Ayarza et al.,
2014; Makris et al., 1985; Tadili et al., 1986; Wigger et al., 1992). These aforementioned results, however,
are in contrast with the high elevation of the Middle Atlas (mean elevation ~2000m),  suggesting that the
mountain chain is isostatically uncompensated at the crustal level (Gomez et al., 1998; Van Den Bosch,
1971), and that the crustal thickness cannot solely account for the high topography (Schwarz and Wigger,
1998). Lithospheric structure was then investigated in order to explain this discrepancy, and the results
show a thin lithosphere beneath the Middle Atlas with a shallow, less than 80km, Lithosphere-
Asthenosphere boundary (Fullea et al., 2010; Teixell et al., 2005; Zeyen et al., 2005). Accordingly, surface
uplift of the Middle Atlas must be a combination of orogenic deformation and asthenospheric processes
(Babault et al., 2008; Clementucci et al., 2023b; Missenard et al., 2006; Pastor et al., 2015; Teixell et al.,



2009). While the first component is well visible in the Cenozoic Middle and High Atlas belts, the second
constitutes a long-wavelength, SW-NE oriented band that affects not only these belts but also the Anti-
Atlas and the Meseta domains (Clementucci et al., 2023a; Missenard et al., 2006; Pastor et al., 2015).
Babault et al., (2008) used scattered stratigraphic paleoelevation markers (e.g uplifted Messinian shallow-
water marine sediments in the Skoura basin, northern Middle Atlas) to infer that the mantle-driven surface
uplift occurred after the Messenian at a rate of ~0.2 mm/yr. Uplift, however, at least in the adjacent
Western Meseta, may have already started in the late Cenozoic as suggested by stratigraphic data and river
profile inversions that yielded uplift rates one order of magnitude lower (~0.05 mm/yr) than those
recorded after the Messinian (Clementucci et al., 2023b).
The Middle Atlas was the site of significant volcanic activity during the Neogene-Quaternary (El
Azzouzi et al., 1999, 2010; Berrahma, 1995; Harmand and Cantagrel, 1984; Rachdi, 1995). This volcanism is
documented by mafic lava flows erupted from a hundred of well-preserved strombolian cones and maars,
forming the Middle Atlas Volcanic Province (MAVP), the largest and youngest volcanic field in Morocco (El
Azzouzi et al., 2010). The MAVP covers an area of ~ 960 km2 essentially within the TMA with scattered
outcrops over the FMA (Fig. 2). El Azzouzi et al. (1999, 2010) carried out a petrologic-and geochemical study
and a systematic K-Ar geochronological dating. This approach allowed to constrain the age (16.25Ma -
0.6Ma) of the lava flows, and to distinguish four lava types, namely Nephelinites, basanites, alkali basalts
and subalkaline basalts. Nephelinites, which constitute 1.2% of the total surface of the MAVP, erupted only
during two events: in the middle to late Miocene and in the Quaternary. The other three lava types are
exclusively Plio-Quaternary in age (El Azzouzi et al., 2010; Harmand and Cantagrel, 1984). The MAVP is part
of the intraplate, alkaline type, Cenozoic volcanism that affected Morocco (El Azzouzi et al., 2010) and  is
located within the elongated zone of thinned lithosphere described above (Missenard et al., 2006). The
Neogene-Quaternary volcanism has contributed to reshape the  landscapes of the Middle Atlas (Amine et
al., 2019; Mountaj et al., 2019). Finally, the fluid lavas flowed through pre-existing valleys for dozens of
kilometres before being incised by the current river streams (e.g., Bou Regreg, Oum Rbia and Sebou Rivers;
Fig. 1).



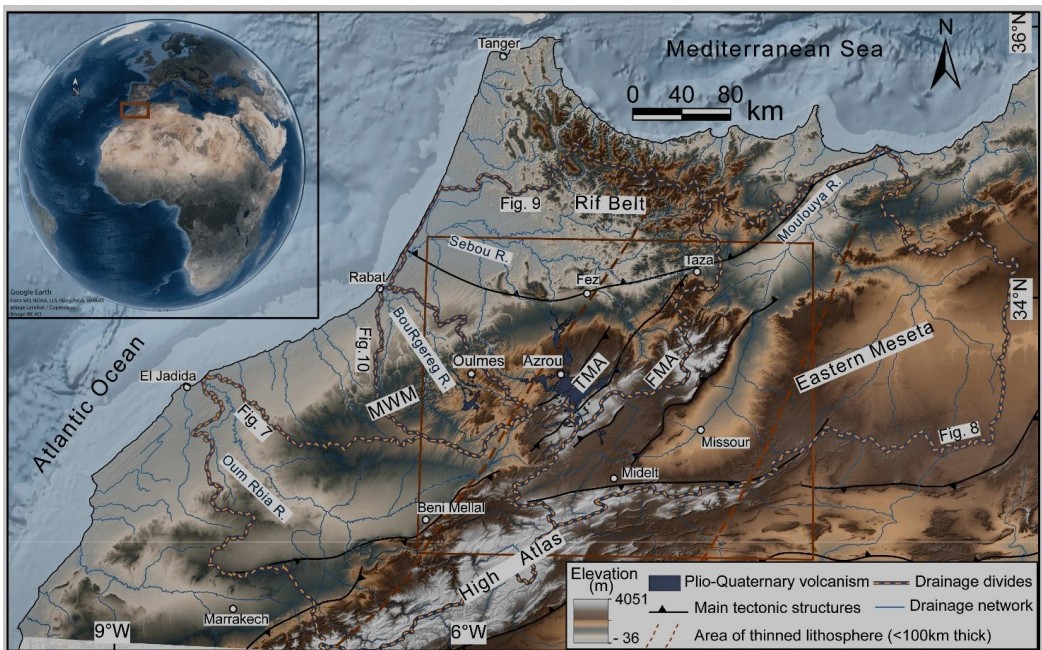

**Figure 1:** Digital Elevation Model (DEM SRTM 90m) of northern Morocco, with the main morphotectonic features. Inset map: Global view from Google Earth (2023). The area of thinned lithosphere is drawn after (Missenard and Cadoux, 2012). The location of figure 2 (Dark brown box), the watersheds of figures 5, 6, 7 and 8 (yellow dotted lines) are indicated. MWM: Moroccan Western Meseta, FMA: Folded Middle Atlas, TMA: Tabular Middle Atlas.

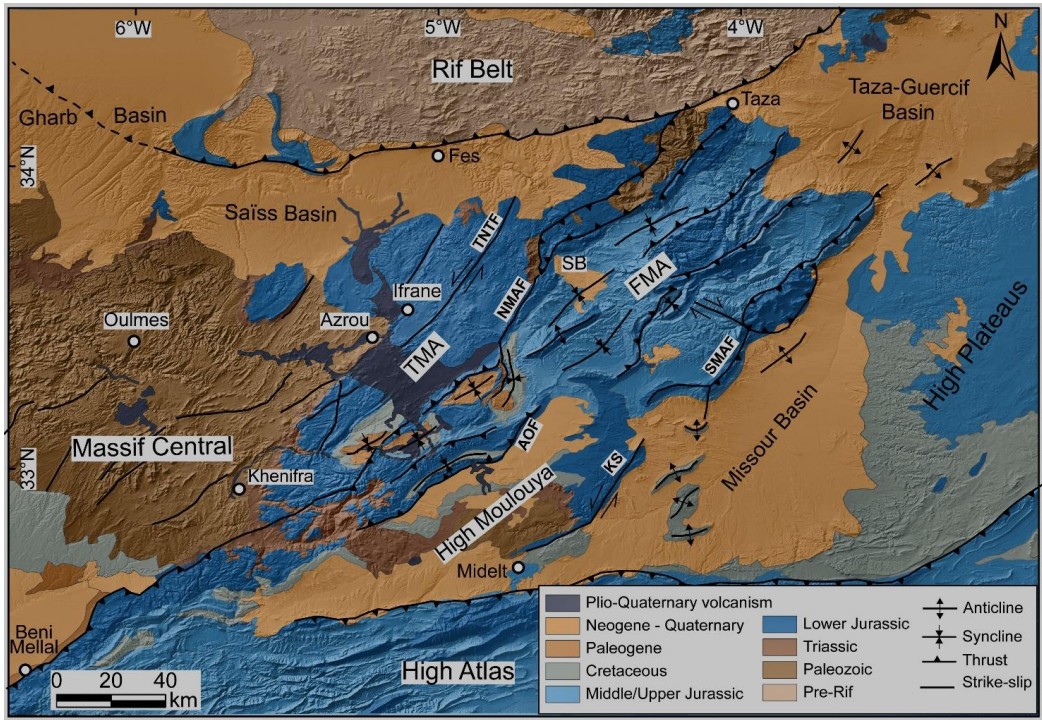




*Figure 2: Simplified geological map of the Middle Atlas and surrounding areas (adopted from the geological map of Morocco (Hollard et al., 1985)). FMA: Folded Middle Atlas, TMA: Tabular Middle Atlas, SB: Skoura Basin, AOF: Ait Oufella Fault, KS: Ksabi Fault, NMAF: North Middle Atlas Fault, SMAF: South Middle Atlas Fault, TNTF: Tizi N'Tghetten Fault.*




Figure 3: Synthetic stratigraphical column of the Middle Atlas (based on Boumir et al. (2023); Charrière (1990); Charrière and Haddoumi (2016); Fedan (1988)).

## 3. Methodology
### 3.1. Topographic analysis
#### 3.1.1 Local relief and swath profiles

Topographic analysis provides first-order information about the degree of incision. To get such
information, we extracted topographic features (local relief maps and swath topographic profiles) from an
SRTM 90m DEM (Downloaded from https://srtm.csi.cgiar.org) using ArcGIS and MATLAB.
Local relief is a parameter describing the maximum dissection of a landscape owing to valley
incision (Kuhni and Pfiffner, 2001). It is defined by the subtraction between maximum and minimum
elevations within a sampling window. We generated the local relief map of the study area in ArcGIS by
means of range statistic in Focal statistics tool using a circular neighborhood of 1 km in radius.
Topographic swath analysis projects elevation values within a rectangular strip perpendicularly to
the midline of the strip. This elevation dataset is used to calculate statistical variables (typically, maximum,
minimum and mean) and to reveal specific characteristics of the topography that cannot be mapped with
one single profile (e.g., Telbisz et al., 2013). The curve of mean elevation gives the general topographic
signal of the landscape. The maximum elevations line marks ridgelines and, when associated with low slope
values, reveals paleo-surfaces. The minimum elevation profile corresponds to the valley floors. The
arithmetic difference between maximum and minimum elevations provides a quick estimation of the
topographic relief. Two swath topographic profiles were produced using the Topographic Analysis Kit (TAK)
(Forte and Whipple, 2019), which is built on the MATLAB-based scripts of Topotoolbox (Schwanghart and
Scherler, 2014). The width of the rectangles, fixed to 10 km, is large enough to enclose the elevation of



both major valleys and ridgelines. The profiles have been vertically exaggerated 10 times to enhance the
topography given the long horizontal distance of the swaths.
3.1.2 Hypsometry

To characterize the stage of geomorphic development of the present-day catchments, we carried
out a hypsometric analysis using the MATLAB functions developed by Jaiswara et al. (2020). The
hypsometric analysis allows studying the distribution of the surface area with respect to the elevation in a
drainage basin (Strahler, 1957). The hypsometric curve is constructed by plotting the relative height (h/H)
against the relative area (a/A). A useful attribute of these curves is that drainage basins of different sizes
can be compared, since area and elevation are plotted as functions of total area and total elevation (Keller
and Pinter, 2002; Pérez-Peña et al., 2009). The hypsometric integral (HI) is defined as the area below the
curve. This area represents the un-eroded volume of the basin. Varying between 0 and 1, the integral of a
given catchment can be calculated as follows:
$$HI = \frac{mean\ elevation - minimum\ elevation}{maximun\ elevation - minimum\ elevation}$$     (1)
The shape of the hypsometric curve and the value of hypsometric integral for a given basin are
related to its degree of dissection and hence to its evolution stage. Convex curves and high values of HI are
indicative of a weakly eroded landscape (youthful stage); S-shaped curves and medium values of HI
characterize moderately eroded regions (maturity stage) while concave curves and low values of HI are
typical of a highly eroded topography (old stage) (Keller and Pinter, 2002; Schumm, 1956; Strahler, 1952).
3.2 River profiles and knickpoints analysis

River longitudinal profiles are plots displaying the variation in channel elevation versus its distance
from the headwaters to the outlet. Modelling a river channel profile is critical to understand its incision
history (Zhong et al., 2022). The stream power incision model (Howard and Kerby, 1983) is the most





commonly used model (Crosby and Whipple, 2006; Whipple and Tucker, 1999) for estimating relative
changes in base-level and hence to quantify the minimum values of fluvial incision and to infer rock uplift
patterns (Ballato et al., 2015; Boulton, 2020; Clementucci et al., 2023b; Lague, 2014)(Ballato et al., 2015;
Boulton, 2020a; Clementucci et al., 2023b; Lague, 2014). The later model defines the variation over time of
the river channel elevation ($dz/dt$) as the difference between the rock uplift (U) and the incision rate (I),
with the incision rate being proportional to upstream drainage area, A, and river gradient, S:
$$\frac{dz}{dt} = U - I = U - KA^m\,S^n \tag{1}$$
Where $K$ is a coefficient of erodibility and $m$, and $n$ are positive constants.
Under steady state condition, the rock uplift rate and incision rate are in dynamic equilibrium,
therefore the channel bed elevation does not change through time ($dz/dt$ = 0). Accordingly, eq. (1)
becomes:
$$S = \left(\frac{U}{K}\right)^{\frac{1}{n}} A^{-\left(\frac{m}{n}\right)} \tag{2}$$
Equation (2) describes a power-law relationship between channel gradient and drainage area, and
is similar to the empirically Flint's equation (Flint, 1974):
$$S = ks\,A^{-(\theta)} \tag{3}$$
Where $k_s$= $(U/K)^{1/n}$ is the channel steepness that is sensitive to changes in rock uplift rate, rock
types, climate,  and $\theta$ = (m/n), which is the concavity index (Snyder et al., 2000; Whipple and Tucker, 1999).
Deviations from this form indicate that the river is in transient state of disequilibrium as a result of base-
level change (Siravo et al., 2021) and geomorphic features (knickpoints) may occur (Boulton, 2020).
From Flint's law (Eq. (3)), one can estimate the parameters $Ks$ and $\theta$ by a linear regression of the
bilogarithmic plot of S and A (Kirby and Whipple, 2012; Wobus et al., 2006). But given the covariance of $Ks$
with $\theta$, it is typical to normalize $Ks$ (becoming $k_{sn}$) using a reference concavity ($\theta_{ref}$), which is commonly set



to 0.45 (Kirby and Whipple, 2012). Moreover, this reference value has been calculated by minimising the
scatter in χ space of all rivers draining the study area by Clementucci et al. (2023b).
We also used the integral approach (Perron and Royden, 2013; Willett et al., 2014), where the
horizontal coordinate (x) of the river profile can be transformed into the χ-coordinate resulting in the
transformed profiles or χ-plots. This method is an alternative to the slope-area analysis, which can be
affected with noise embedded in DEMs.
By integrating Eq. (2) and considering U and K as constants in space and time, Eq. (4) can be
obtained:
$$z(x) = z(x_b) + k_s\, A_0^{-\left(\frac{m}{n}\right)} \chi \qquad (4)$$

With   $\chi = \int_{x_b}^{x} \left(\frac{A_0}{A(x)}\right)^{\frac{m}{n}} dx$   (5)
Where $x$ is the upstream distance from the outlet $x_b$, $A$ is the drainage area and $A_0$ is a reference area.
We extracted the longitudinal river profiles, χ-plots, and the $K_{sn}$ map using the ChiProfiler (Gallen
and Wegmann, 2017). We only focused on river channels where incision rates from lava flows were
estimated. The parameters were set as follows: $\theta_{ref}$ = 0.45, $A_0$ = 1 m$^2$ and $A_{cri}$ = 10$^6$ m$^2$ (Critical drainage area,
a threshold drainage area).
In order to classify the knickpoints, we checked them out using geological maps and Google Earth
satellite images. Therefore, we labelled them as minor, lithological, and non-lithological. Minor knickpoints
(blue circles) are those coincident with dams, and do not show significant variations in $K_{sn}$ upstream and
downstream or are just artifacts. Lithological knickpoints (yellow circles) are located between two
lithologies with different rock strengths. Non-lithological knickpoints (Dark brown circles) are associated
with important slope break in the profile and prominent low-gradient reach upstream.
**3.3 Incision rates**



The geomorphic relationship between river channels and Quaternary lavas flows in the Middle
Atlas and the Moroccan Massif Central offers a good opportunity to quantify the fluvial incision rates.
During their emplacement, the lava first flowed down the river channels (Fig. 4a) and then, once cooled,
were incised by the same rivers. Following lava emplacement, if there is no substantial change in boundary
conditions, such as a base-level drops and/or changes in precipitation regimes, the river would likely
continue incising until it reaches its previous position marked by the lava-substrate stratigraphic limit (Fig.
4b). Otherwise, the fluvial incision would extend beyond that limit to adjust to any imposed perturbation
(Fig. 4c). Dividing the current valley depth measured from the top of the lava flow by the age of the lava
gives the rate of post-lava emplacement incision. In our study, valley depths were measured in the field
using a GPS device (GARMIN GPS72H) and where suitable a meter stick, while lava flow ages were taken
from available radiometric ages (El Azzouzi et al., 2010; Rachdi, 1995).

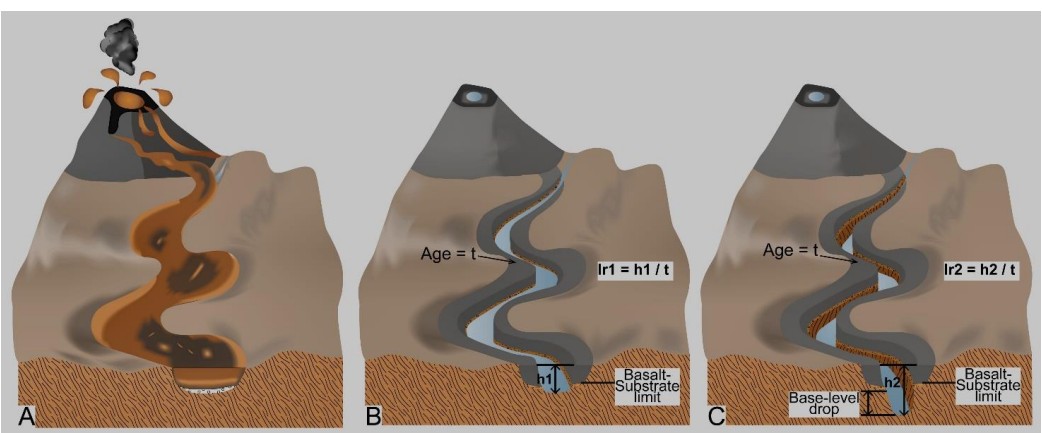

**Figure 4: Schematic 3D diagrams depicting lava-rivers interactions. A) After the emplacement of a lava flow in a pre-existing river valley, the river will start incising the cooled lava until it attains its original base-level (B) it extends beyond the lava-substrate boundary in response to a prolonged base-level fall (C).**




## 4. Results

### 4.1. Local relief and swath profiles

The local relief map (Fig. 5A) shows the highest values (480-1100m) in the Folded Middle Atlas (Between the NMAF and the SMAF), especially in its northeastern part near the major faults. The Tabular Middle Atlas exhibits mostly low values of local relief, that tend to slightly increase near the Tizi N'Teghtten and Dayet Aoua faults. The lowest values of local relief are found in the bordering Saïs sedimentary basin to the northwest, in the Taza-Guercif basin to the northeast and in the Missour basin to the southeast. The Massif Central shows an inhomogeneous distribution of local relief with values ranging from 270 to 580 m around Oulmes, that tend to decrease to the East along the corridor running between Azrou and Khenifra where they reach values of less than 270 m. While the lava flows exposed in the Bou Lahmayel valley (BLV) present relatively high local relief, ranging from 270 m to 580 m, the surface of the lava flows typically displays comparatively lower values, varying from 0 to 170 m. Nevertheless, the lava flows can be surrounded by zones of relatively high relief such as in the Guigou valley (GV) and southwest of Azrou where they range from 400 m to 580 m (Fig. 5A). The outlines of some river valleys are delineated by a noticeable contrast of local relief reflecting their high degree of dissection (e.g., Cheg El Ard Valley (CAV); Fig. 5A).

The topographic swath profiles (SW 1 and SW 2; Fig. 5B and C) are oriented WNW-ESE, perpendicular to the main structures of the Middle Atlas and surroundings and highlight the characteristics of the different morpho-structural domains of the study area. In the SW 1 swath, the Folded Middle Atlas shows high relief and consequently high fluvial incision (> 700 m). In contrast, the Tabular Middle Atlas (TMA), where most of the lava flows are located, represents an elevated low relief area with topographic relief not exceeding 300 m, while in the Saïss and Missour basins the maximum and minimum topography curves are very close indicating very low topographic relief. In the SW 2 swath, which runs from the Western Meseta to the High Moulouya and the Missour basin, the difference in the degree of incision



between the FMA and TMA is smoothed. The High Moulouya also exhibits a flat topography and a general
low incision except for the relief associated with the Ksabi fault. In contrast, the Central Massif shows high
relief and a topographic relief of about 500 m in the first 20 km of the profile and low topographic relief in
the second half of the profile.

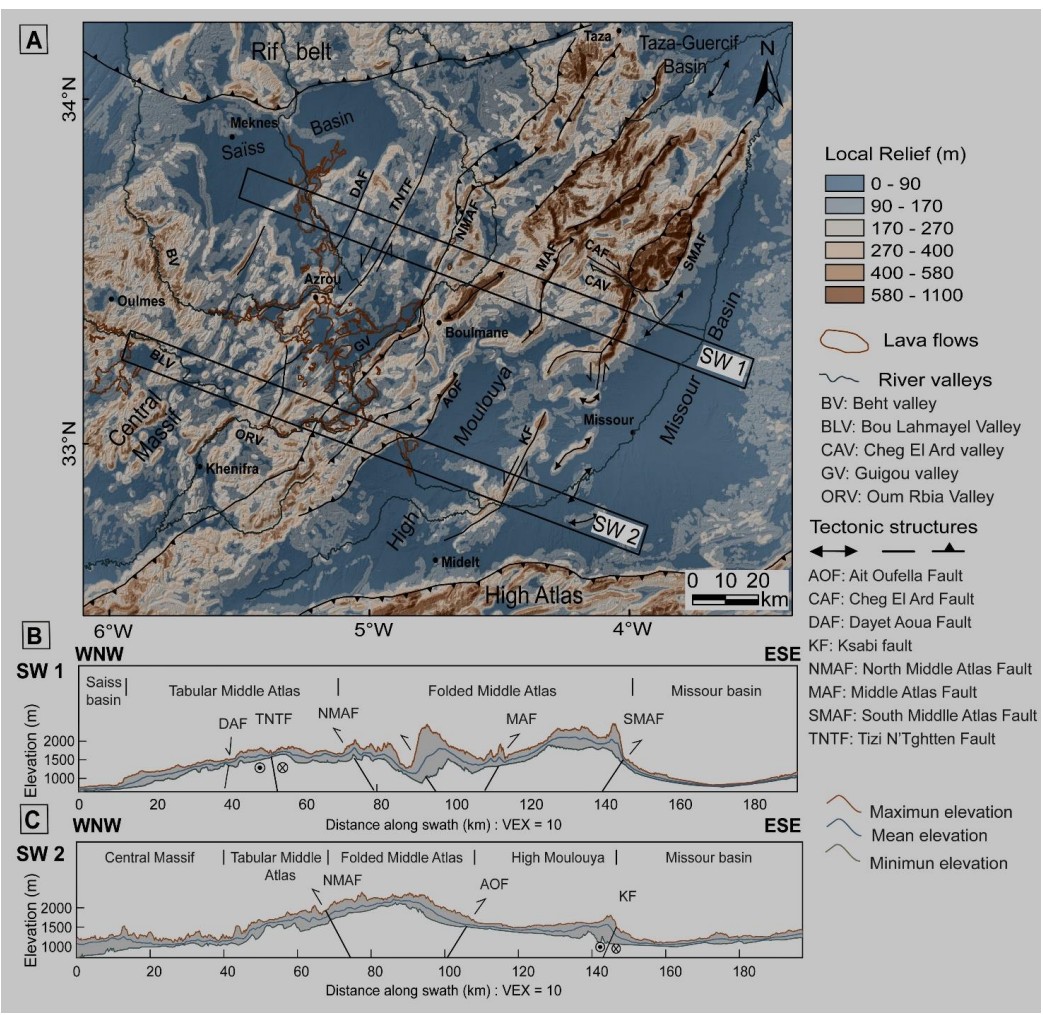

**Figure 5: A) Local relief map of the Middle Atlas and surroundings with a 2 km moving window. B and C) Topographic swath profiles perpendicular to the main structures of the study area with their locations indicated in A.**





### 4.2. Hypsometric and fluvial analysis

The drainage network in the Middle Atlas and surrounding plateaus includes four main rivers and
related tributaries (Fig. 1). Three rivers, Oum Rbia, Bou Regreg and Sebou, are located on the western side
of the belt and drain to the Atlantic Ocean. The Oum Rbia and Sebou rivers originate in the Middle Atlas
while the Bouregreg originates in the Western Meseta. One river, Moulouya, is located on the eastern side
of the belt and drains to the Alboran sea (western Mediterranean).
Notable variations in channel steepness are observed over the drainage network of the studied
area (Fig. 6). The *Ksn* map shows generally higher values in the northeastern part of the Middle Atlas,
especially in the river channels draining the NE fault-bounded front of the belt (SMAF). These high *Ksn*
values are consistent with high local relief (Fig. 5A). The lowest values of *Ksn* are observed in the TMA, in
the High Moulouya, and in the eastern part of the Massif Central. The river valleys where lava flows were

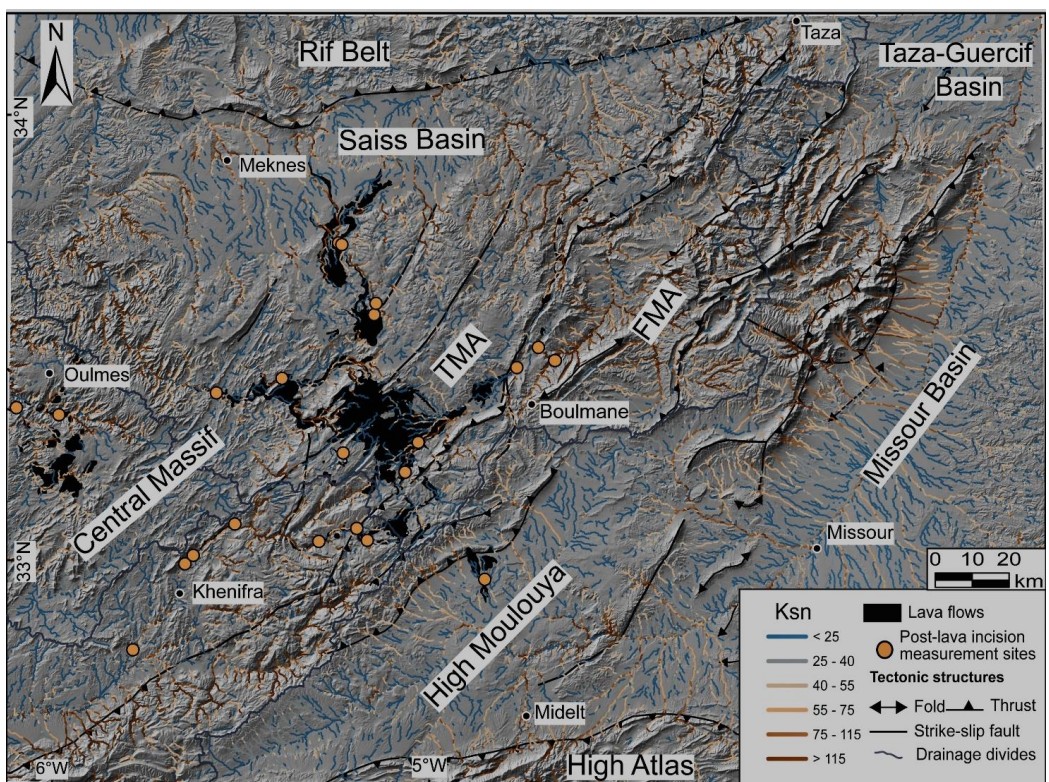

**Figure 6: Normalized channel steepness indices (Ksn) of the Middle Atlas and surroundings, superimposed on a hillshade relief map.**



emplaced generally exhibit low values of *Ksn*. However, it is noteworthy that the *Ksn* values are relatively
higher in lava-bearing channels in the Massif Central south of Oulmès, than in the Middle Atlas.

### 4.2.1. Oum Rbia catchment

The catchment of Oued Oum Rbia has a W-E elongated shape with a drainage area of about 38000
km$^2$ and drains the southwestern termination of the Middle Atlas, the northwestern slopes of the High
Atlas, and the southern slopes of the phosphate plateaus (Fig. 7A). The Oum Rbia River originates from the
Oum Rbia springs which are located in the Middle Atlas domain. The trunk river has a length of 555 km
from the springs to the outlet in the Atlantic Ocean. The main tributaries are those of its left bank namely
Serrou, El Abid, and Tassaouat rivers. The trunk channel has a W-E trend in the Middle Altas, then it
deviates to a NE-SW trend immediately at the exit of the Middle Atlas until the Tadla plain where the trend
changes to a mean NW-SE direction.
The hypsometric curve of the Oum Rbia catchment is concave with a value of the hypsometric
integral of 0.26 (Fig. 7C). Its longitudinal profile displays a series of knickpoints with different origins (Fig.
7D, E). The knickpoint located on the North Middle Atlas Fault is the most important one (Fig. 7B, D, E)
since it depicts an abrupt change in the slope with a flat segment upstream (Fig. 5E). The lithological
knickpoint (yellow circle) matches the transition between Eocene and Cretaceous strata. The Knickpoint
(dark brown circle) just downstream of the lithological knickpoint is located upstream of the Oum Rbia
sources and Mermel Fault (Fig. 7B). The knickpoints in blue correspond to dams.







Figure 7: A) Simplified geological map of the Oum Rbia catchment, B) Close up view of upper course of the Oum Rbia with values of post-lava emplacement incision rates reported, C) Hypsometric curve and Integral index of the Oum Rbia catchment, and D) Longitudinal and E) Chi profile of the Oum Rbia River.




### 4.2.2. Moulouya catchment

The Moulouya drainage basin covers an area of about 60408 km$^2$ and drains the southeastern flank of the Middle Atlas, the northern flank of the High Atlas, the western slopes of the high plateaus, and the eastern Rif (Fig. 8A). The trunk channel is about 600 km long and trends SW-NE. Its sources are in the junction between the High and the Middle Atlas and its outlet is in the Mediterranean Sea.

The hypsometric analysis of the Moulouya catchment reveals an S-shaped hypsometric curve with a hypsometric integral value of 0.31 (Fig. 8B). Pastor et al. (2015) provided a detailed river profile and knickpoint analysis of the Moulouya River and its main tributaries. Here we focus only on the lava filled Tanfit River and, for comparison, the Cheg El Ard River where Pastor et al. (2015) calculated incision rates using fluvial terraces. The Tanfit channel shows a knickpoint (dark brown circle) located on the Paleozoic basement (Fig. 8A, C, and D) consistent with a local increase in the channel steepness (Fig. 6). The Moulouya River exhibits a minor knickpoint upstream of the Ksabi Fault (blue circle) and a non-lithological knickpoint in the Beni Snassen massif at the lower course of the river (Fig. 8A, C, and D). The Cheg El Ard River flows over the active southeastern deformational front of the Middle Atlas and coincides with the Cheg El Ard dextral fault CAV (Laville et al., 2007)). The Cheg El Ard longitudinal profile exhibits tree knickpoints (Fig. 8A, C, D). The uppermost knickpoint is prominent and well expressed in the Chi-plot (Fig. 8D) with a low-gradient reach upstream and a steep segment downstream. The knickpoint (blue circle) does not reveal a significant change in channel steepness upstream and downstream. The lowermost knickpoint coincides with the active Beni Aioun Anticline (BBA; Pastor et al., 2015).





**Figure 8: A) Simplified geological map, B) hypsometric curve and Integral index of the Moulouya catchment, C) Longitudinal and D) Chi profile of the Cheg El Ard and the Tanfit/Moulouya rivers.**



### 4.2.3. Sebou catchment

The Sebou catchment has a roughly hexagonal shape and covers an area of 38718 km². The Sebou River drains the northern slopes of the Western Meseta, the northwestern slopes of the Middle Atlas, and the southern slopes of the Rif belt (Fig. 9A). The Sebou River originates from the Middle Atlas and flows into the Atlantic Ocean near Kenitra (Fig. 9A). The trunk channel is about 520 km long and flows towards the NW in the upper and middle course then it turns towards the West in the lower course. The Sebou River as well as its Middle Atlas tributaries flow through the Saïss Basin and cross over the Rifain frontal thrust before turning toward the Atlantic. The Sebou drainage basin shows a concave hypsometric curve and a hypsometric integral value of 0.22 (Fig. 9B). The concavity of the curve is related to the presence of the large Gharb basin in the lower part of Sebou catchment (Barcos et al., 2014).

The river longitudinal profiles of the three lava-filled channels present some knickpoints and knickzones that denote disequilibrium (Fig. 9C and D). The Sebou-Guigou river profile shows a 1200 m-high lithological knickpoint coinciding with the contact between the dolomitic and calcareous layers of the lower Jurassic and the marly limestone layers of the middle and upper Jurassic. A non-lithological knickpoint is identified in the Sebou River upstream the North Middle Atlas Fault (Fig. 9A, B, C). The longitudinal profile of Tizguite River exhibits a lithological knickpoint in the lower river course, at the end of lava flow, and a minor knickpoint in the upper reach. The river profile of Beht-Tigrigra displays a 500m-high knickpoint at about 250 km from the outlet (Fig. 9C). Downstream this knickpoint, the channel has entrenched deeply (Fig. 9A), and its steepness increases (Fig. 6). Clementucci et al. (2023b) provided data of denudation rates from [10]Be cosmogenic isotopes for the Beht-Tigrigra tributary showing values ranging between 19.5 to 40 m Ma$^{-1}$ for the low-relief landscape (upstream of the knickpoint) and high-relief topography (downstream of the knickpoint), respectively.





**Figure 9: A) Simplified geological map, B) hypsometric curve and Integral index of the Sebou catchment, C) Longitudinal and D) Chi profile of the Sebou-Guigou, Beht-Tigrigra and Tizguite rivers.**




### 4.2.4. Bou Regreg catchment


The Bou Regreg catchment is located within the western side of the Middle Atlas and has a fan-like
shape with an area of 9834 km2. The river system drains the Paleozoic units of the Massif Central, which
include from East to West: The Kasbat Tadla-Azrou anticlinorium, Fourhal-Tilt synclinorium, Khouribga-
Oulmes anticlinorium, and the synclinorium of Romani in the lower course (Fig. 10A). The trunk channel has
a mean NW-SE trend, and a length of ca. 250 km from its mouth in the Atlantic Ocean to its headwaters.
Unlike the other rivers, the Bou Regreg does not reach the Middle Atlas. The hypsometric curve of the Bou
Regreg catchment shows an S-shaped curve with a hypsometric integral value of 0.42, higher than those of
the previously described watersheds (Fig. 10B).
The longitudinal profile of the lava-filled rivers (Bou Lahmayel-Bou Regreg and Ksiksou) are not
concave-up profiles typical of an equilibrium state, but they show knickpoints highlighting the occurrence
of some perturbations. The knickpoints, noted by the Dark brown circles, are 1000 m high and show
prominent inflection in the profiles (Fig. 10C and D). The knickpoints assigned as minor do not present
prominent slope break and in some cases, they may be artifacts. Here, basin-wide denudation rates range
between 19.8 and 17.4 m Ma$^{-1}$ for Bou Lahmayel-Bou Regreg and Ksiksou, respectively (Clementucci et al.,
2023b).





**Figure 10: A) Simplified geological map, B) Hypsometric curve and Integral index of Bou Regreg catchment, C) Longitudinal and D) Chi profile of Bou Lahmayel and Ksikssou rivers.**




### 4.3. Post-eruption incision

We estimated the post-lava emplacement incision rates in twenty-two sites distributed over the different morpho-structural domains of the studied area, namely, the Moroccan Massif Central, the Tabular Middle Atlas, and the Folded Middle Atlas. The measurements are scattered over the river valleys occupied by the lava flows (Fig. 11A; Table 1), which are distributed along the main rivers that drain the Middle Atlas and the surrounding plateaus.





Figure 11: A) Digital topography of the study area showing the post-lava incision rates, B) Average annual rainfall map from 1901 to 1981 using the data "CRU TS Version 4.06" from the Climatic Research Unit (University of East Anglia) and the National Centre for Atmospheric Science (NCAS) (https://crudata.uea.ac.uk/cru/data/hrg/ ). The locations of figures 12 and 13 are indicated.



**Table 1: Summery of local post-lava fluvial incision rates in the Massif Central and the Middle Atlas. Lava ages are from El**

**Azzouzi et al., (2010).**

| No. | Location | GPS Coordinates | | Local post-lava incision (m) | Lava age (Ma) | Post-lava incision rates (mm/yr) | Rock type | Substratum rock type |
|---|---|---|---|---|---|---|---|---|
| | | Longitude | Latitude | | | | | |
| 1 | Aourach bridge | -5.78915 | 32.80308 | 30 | 1.65 ± 0.08 | 0.02 | Basanites | Schists |
| 2 | 5km N of Khenifra | -5.6527 | 32.99626 | 13 | 1.65 ± 0.08 | 0.01 | | Not reached |
| 3 | El Borj | -5.63086 | 33.01562 | 20 | 1.65 ± 0.08 | 0.01 | | Not visible |
| 4 | Tanafnite | -5.52048 | 33.08681 | 14 | 1.65 ± 0.08 | 0.01 | | Not reached |
| 5 | Fellate | -5.29624 | 33.04903 | 36 | 1.65 ± 0.08 | 0.02 | | Limestones |
| 6 | Bekrit canyon | -5.19539 | 33.07887 | 30 | 1.65 ± 0.08 | 0.02 | | Limestones |
| 7 | Bou Anguar | -5.16721 | 33.05215 | 25 | 1.65 ± 0.08 | 0.02 | | Not reached |
| 8 | Am Larais | -4.85355 | 32.96532 | 22 | 1.56 ± 0.08 | 0.01 | | Limestones |
| 9 | Ait Youssef | -5.39753 | 33.415 | 23 | 2.19 ± 0.18 | 0.01 | Alkali basalts | Not reached |





| | | | | | | | | |
|---|---|---|---|---|---|---|---|---|
| 10 | Amghas | -5.57293 | 33.3812 | 32 | 2.19 ± 0.18 | 0.01 | | Schists |
| 11 | Outgui 1 | -5.24838 | 33.71991 | 37 | 1.05 ± 0.26 | 0.04 | | Dolostones |
| 12 | Outgui 2 | -5.23958 | 33.71626 | 13 | 0.89 ± 0.31 | 0.01 | | |
| 13 | Ait Qessou | -5.06698 | 33.20561 | 30 | 0.85 ± 0.07 | 0.04 | | Not reached |
| 14 | Ait Said Ouhaddou | -5.03209 | 33.27284 | 20 | 2.41 ± 0.28 | 0.01 | | Not reached |
| 15 | Charchara | -4.76885 | 33.44135 | 32 | 2.58 ± 0.12 | 0.01 | | Conglomerats |
| 16 | Taferdouste | -4.67204 | 33.45911 | 21 | 2.58 ± 0.12 | 0.01 | | Marls |
| 17 | Taghazoute | -4.71334 | 33.48271 | 19 | 2.58 ± 0.12 | 0.01 | | Limestones |
| 18 | Tizguit | -5.15165 | 33.55932 | 17 | 1.14 ± 0.11 | 0.01 | Subalkaline basalts | Dolostones |
| 19 | Zaouia d'Ifrane | -5.14672 | 33.58471 | 9 | 1.14 ± 0.11 | 0.01 | | |
| 20 | Ain Kahla | -5.23267 | 33.24819 | 38 | 2.33 ± 0.1 | 0.02 | Nephelinites | Not reached |
| 21 | Bou Tsaggatine | -5.99678 | 33.3289 | 122 | 0.93 ± 0.07 | 0.13 | Tephrite | Metaturbidites |
| 22 | Bled | -6.107813 | 33.34059 | 110 | 1.1 ± | 0.10 | Basanites | |



| | | | | | | |
|---|---|---|---|---|---|---|
| Belmadi | | | 0.1 | | | |

### 4.3.1. Moulouya catchment

The lava flows used in this study within the Moulouya watershed erupted from the Am Larais volcano (Fig. 8A). The lavas flowed for about 14 km along the Tanfit River, a tributary of the Moulouya River. These volcanic rocks are basanites and have a K/Ar radiometric age of $1.56 \pm 0.08$ Ma (El Azzouzi et al., 2010) . Our field observations show that these Quaternary lavas cross the Ait Oufella fault (Fig. 6A). The post-lava emplacement incision depth, undertaken upstream of the non-lithological knickpoint in the Tanfit River (Fig. 8A and D), is about 22 m, which corresponds to an incision rate of 0.01 mm yr$^{-1}$.

### 4.3.2. Oum Rbia catchment

The lava flows in the Oum Rbia River erupted from the Tamarrakoït volcano (Martin, 1981). These lavas are also basanites and were dated in two localities (El Azzouzi et al., 1999; 2010). In El Borj (5 km North of Khenifra) they yielded a K/Ar age of $1.01 \pm 0.04$ Ma and near the Bekrite village they yielded a K/Ar age of $1.65 \pm 0.08$ Ma (Fig. 7B). The lavas flowed for about 90 Km from the Middle Atlas to the Western Meseta along the Oum Rbia River and its upper tributaries (Fellate and Amengous Rivers; Fig. 7B). Our field observations, indicate that these Quaternary basanites cross the NMAF. The incision rate into the Quaternary lava was measured in seven stations along the river upstream and downstream of knickpoints. The magnitude of fluvial incision of the lavas varies from 13 to 36 m indicating incision rates between 0.01 and 0.02 mm yr$^{-1}$.

### 4.3.3. Sebou catchment

The majority of the lava flows of the studied area is circumscribed within the Sebou watershed (Fig. 9A). They erupted from the volcanic centres distributed over the tabular Middle Atlas and flowed along pre-existing valleys to the plains of the hinterland (Martin, 1981). The lavas flowed mainly along the Guigou, Tizguit and Tigrigra tributaries. Along the Guigou channel, alkali basalt lava flows that travelled for about 40 km to eventually stop in the Skoura Basin, yielded K/Ar ages of $2.41 \pm 0.28$ and $2.58 \pm 0.12$ Ma (El Azzouzi



et al., 2010). Along the Tizguite channel, alkaline and subalkaline lavas were emitted from two main
volcanoes (Elkoudiat and Outgui) and flowed for a cumulative length of about 35 km. The lavas erupted
from the Outgui Volcano are alkali basalts with a mean K/Ar age of 1.61 Ma, while the lavas from the El
Koudiat volcano are subalkaline basalts  and have a mean K/Ar age of 1.05 Ma (El Azzouzi et al., 2010). The
lavas also flowed from the Tabular Middle Atlas towards the lowland of the Massif Central along the
Tigrigra valley and other steep ravines for about 50 km. The volcanics used to measure incision rates in the
Tigrigra valley, are alkali basalts and yielded a K/Ar age of 2.19 ± 0.12 Ma (MA412 sample of El Azzouzi et
al., 2010). Our field observations indicate that lavas in the Tabular Middle Atlas cross the Tizi N'Tghetten
strike-slip Fault without being deformed by the fault (TNTF; Fig. 9A). The vertical incision into the lavas
varies from 9 to 38 m which, together with the K/Ar ages, indicate post lava incision rates between 0.01
and 0.04 mm yr$^{-1}$.

### 443    4.3.4. Bou Regreg catchment

The lava flows encompassed in the Bou Regreg catchment erupted from about twenty volcanoes.
The lavas flowed for about twenty kilometers along the Bou Lahmayel and Ksiksou Rivers, which are
tributaries of the Bou Regreg River (Fig. 10). The lava flows, we have used for the incision rate calculation,
are tephrites in the Bou Lahmayel River and basanites in the Ksikssou River (Baudin et al., 2001; Bouhdadi
et al., 2002). Their K/Ar ages are 0.93 ± 0.07 and 1.10 ± 0.10 Ma respectively (Rachdi, 1995). The vertical
incisions into the lava flows are 122 m in the Bou Lahmayel River and 110 m in the Ksiksou River. Combined
with the K/Ar ages, they indicate incision rates of 0.13 and 0.10 mm yr$^{-1}$ respectively. Both measurements
are located downstream of the non-lithological knickpoints (Fig. 10A and C). It noteworthy that the post-
lava emplacement fluvial incision measured in the Bou Regreg catchment goes several tens of meters
beyond the lava-substrate interface (Fig. 12B and 13A).
To summarize, the magnitude of post-lava emplacement incision varies between 9 and 38 m in
twenty measurement sites (Sites N°1 to N°20), while it reaches 110 m and 122 m in two sites located in the
Bou Regreg catchment (Table 1; N°21 and 22). The lava flows are Quaternary with radiometric ages varying



between 2.58 and 0.85 Ma. The calculated incision rates range between 0.01 and 0.13 mm yr$^{-1}$. The incision
rates are not correlated with the ages of the lava flows. The incision rates calculated in the Bou Regreg
catchment (0.1-0.13 mm yr$^{-1}$) are one order of magnitude higher than those calculated in the other river
catchments, as also shown by available basin-wide erosion rates for the same region averaged over a time
scale of few tens of thousands of years (Clementucci et al., 2023b).
## 5. Discussion

The 2.58- to 0.85 Ma-aged lavas exposed along the valleys of the Middle Atlas and the Massif Central

allow us to calculate post-lava emplacement fluvial incision rates. Combined with topographic and fluvial
analysis, this dataset help constraining the evolution of the landscape at regional scale (Middle Atlas and
surroundings) over the Quaternary. Overall, we observe one order of magnitude difference in incision rates
between the Western Meseta and Middle Atlas. The Massif Central shows relatively high post-Quaternary
incision rates in the order of 0.1 mm yr$^{-1}$, while the other locations present rather uniform incision rates of
about 0.02 mm yr$^{-1}$. Our geomorphic analysis is consistent with this dichotomy in fluvial incision rates
between the Bou Regreg catchment and the other catchments. As such, higher local relief and hypsometric
index values are relatively higher in the Massif Central sector with respect to the Middle Atlas domain.

Many studies have interpreted incision rates in terms of rock uplift (Clementucci et al., 2023b; Evenstar

et al., 2020; Lavé and Avouac, 2001; Ott et al., 2018; Pan et al., 2013; Pastor et al., 2015; Wu et al., 2020),
however, there are other factors, such as climate and lithology that can influence the rate of incision
(Campforts et al., 2020; Clementucci et al., 2022). Below we discuss the effect that each of these three
mechanisms may have had on the observed spatial variations of incision rates between the Western
Meseta and the Middle Atlas.

### 5.1. Lithology





Bedrock is known to exert a significant impact on the rate of erosion (Clementucci et al., 2022;
Montgomery and Gran, 2001; Moumeni et al., 2024; Sklar and Dietrich, 2001; Zondervan et al., 2020). Due
to differences in mineral composition and chemical structure, different rock types have varying levels of
resistance to erosion (Burbank and Anderson, 2011), and less-resistant lithologies tend to be incised more
easily than high-resistant lithologies (e.g. Clementucci et al., 2022; Zondervan et al., 2020). Rock-strength
data from the adjacent High Atlas indicate that Paleozoic meta-sediments and Mesozoic limestones have
similar Uniaxial Compressive Strength UCS, with average values of 55 and 57 MPa, respectively (Zondervan
et al., 2020). Both lithologies are widely exposed in the Massif Central and Middle Atlas and hence
constitute the substrates for lava flows. Nevertheless, our estimated incision rates show one order of
magnitude difference between the Middle Atlas and the Western Meseta (table 1). This observation
suggests that lithology cannot fully account for the observed spatial variations in incision rates, at least on a
Quaternary time scale.

5.2. Climate/Precipitation

Climate is another major factor that controls surface processes and thus erosion rates (England and
Molnar, 1990; Ferrier et al., 2013; Hartshorn et al., 2002; Willett et al., 2001). An increase in precipitation,
and thus in rivers runoff, enhances the capacity of rivers to incise (Lin et al., 2021; Pérez-Peña et al., 2009;
Whipple, 2009). Analogue and numerical modelling also suggest that a gradient in precipitation can lead to
differential erosion rates and then to divide migration towards the drier side of an orogen (Bonnet, 2009;
Reitano et al., 2023; Willett, 1999). To investigate the potential impact of climate on erosion rates, many
studies used modern precipitation rates as proxy for climatic forcing (e..g., Champagnac et al., 2012;
Palumbo et al., 2010; Pan et al., 2010; Struth et al., 2020; Winiger et al., 2005; Wu et al., 2020). Figure 11B
shows the average annual precipitation map of the study area during the period 1981-2011 interpolated
using data from the Climatic Research Unit (University of East Anglia) and the Met Office (Harris et al.,
2020). The rainfall map does not reveal any significant contrast in precipitation between the Bou Regreg



catchment, in which we measured higher incision rates, and its surrounding watersheds. The highest
precipitation, around 800 mm yr$^{-1}$, falls in the triple junction of the Bou Regreg, Oum Rbia, and Sebou
drainage divides. In addition, most of the Moulouya catchment receives lower precipitation in comparison
with the other catchments, although data from Pastor et al., (2015) showed high fluvial incision rates in the
Cheg El Ard valley (Fig. 8). Therefore, based on the precipitation pattern represented in figure 11B and
assuming that this distribution remained similar during the Quaternary, the rainfall spatial gradient alone
cannot explain the spatial variation of our fluvial incision rates. The apparent independence of basin-wide
denudation rates on precipitation is also testified in the recent work conducted by Clementucci et al.
(2023b) in the same study area using IMERG data.

**5.3. Surface uplift**

Given that lithology and climate fail to account for the spatial variation in fluvial incision rates, we

explore rock uplift induced by tectonic, and mantle-driven processes as potential cause. The Moroccan
Massif Central constitutes the largest massif of the Paleozoic Western Meseta realm. These terrains consist
of metamorphic and magmatic rocks that were assembled during the Variscan orogeny (Michard et al.,
2023). Previous studies characterized this region as tectonically stable following the Hercynian orogeny
(e.g, Michard, 1976). However, recent studies have provided new insights, revealing that this area has
undergone vertical movement during the Mezo-Cenozoic times (Barbero et al., 2011; Clementucci et al.,
2023b; Ghorbal et al., 2008; Yaaqoub and Essaifi, 2023). On the other hand, the Middle Atlas is a Cenozoic
intracontinental belt that resulted from the tectonic inversion of a Triassic-Jurassic rift that once affected
the Moroccan Meseta (Charrière, 1990). The tectonic inversion of the Middle Atlas began in the late
Cretaceous and culminated during the Oligocene-Neogene (Charrière, 1990). The uplift of the Middle Atlas
is the result of the superposition of mantle-driven uplift and tectonic uplift related to the Cenozoic
shortening (Babault et al., 2008; Clementucci et al., 2023b; Lanari et al., 2023; Pastor et al., 2015).



The inferred Quaternary incision rates from the lava flows in the Middle Atlas are one order of
magnitude lower than the estimates for the Moroccan Central Massif. The analysis of geomorphic indices
such as local relief (Fig. 5) and channel steepness ($k_{sn}$; Fig. 6), which is a proxy for rock uplift assuming
similar rock erodibility and precipitation rates and also in case of equilibrated river profiles (e.g., Ahnert,
1970; Kirby and Whipple, 2012; Safran et al., 2005; Wu et al., 2020), mimics the distribution of the incision
rates with lower values that are typically observed in the Middle Atlas. Moreover, the hypsometric analysis
of the Bou Regreg river reveals a high hypsometric index (Fig. 10B) suggesting that the Bou Regreg
catchment is still in a young stage of geomorphic evolution (e.g, Barbero et al., 2011). This geomorphic
immaturity is also illustrated by the river longitudinal profiles of the lava-filled valleys (Fig. 10 C and D),
which exhibit elevated non-lithological knickpoints indicating topographic transience most likely triggered
by changes in surface uplift rates (e.g., Clementucci et al., 2023b).
Our Quaternary incision rates in the Bou Regreg watershed, draining the Moroccan Massif Central, are
in the order of 0.1 mm yr$^{-1}$. These rates are comparable in magnitude to those presented by Pastor et al.
(2015) for the last 411 ka and the uplift rates, averaged over the last 5 Ma, estimated by Babault et al.
(2008). Pastor et al. (2015) combined fluvial geomorphic analysis and incision rates from Quaternary river
terraces in the Cheg El Ard valley to infer the recent tectono-geomorphic evolution of the Missour basin
and the surrounding Middle Atlas and High Plateaus hinterlands. The occurrence of incised and
undeformed fluvial terraces in the tectonically quiescent distal part of the Missour Basin was interpreted to
reflect a regional dynamic topography with uplift rates of 0.1-0.2 mm yr$^{-1}$ (Babault et al., 2008, Pastor et al.,
2015). Conversely, the fluvial incision rates value of 0.44 mm yr$^{-1}$, calculated near the Middle Atlas front
(Fig. 8), in an area affected by recent fault-propagation folds, indicate incision related to the combined
effect of regional dynamic uplift (≈0.1-0.2 mm yr$^{-1}$) and local thrust-related uplift (≈0.3 mm yr$^{-1}$). Thus, the
main conclusion of these studies was that the regional surface uplift results from the combination of
mantle-driven surface uplift and structural surface uplift produced by the frontal structures of the Middle
Atlas. Given that there is no evidence indicating that the Moroccan Massif Central was affected by Neogene



tectonic shortening (Barbero et al., 2011), the post-lava emplacement incision rates we measured must be
related to mantle-related surface uplift.

On the other hand, the fluvial incision rates measured from the lava flows in the watersheds that drain

the Middle Atlas Mountain belt (Fig. 7, 8, 9) are in the order of 0.01 mm yr$^{-1}$. These values are significantly
lower than those related to the tectonic uplift reported in the eastern flank of Middle Atlas by Pastor et al.
(2015). It is worth noting that the eastern flank of the Middle Atlas where the thrust-related (≈0.3 mm yr$^{-1}$)
incision rates were calculated, is bounded by the SMAF, which is an active complex fault zone, where
Jurassic carbonates are thrusted onto coarse-grained deposits of the Neogene-Quaternary Missour basin
(Delcaillau et al., 2007; Laville et al., 2007). Therefore, the Cheg El Ard River, which drains the tectonically
active Jbel Bou Nacer (the highest summit in the Middle Atlas) and crosses the active fault-related folds
associated with the SMAF, is such a powerful river that has entrenched deep and narrow canyons (Fig.
13D). In contrast, the western flank of Middle Atlas, where we calculated the majority of incision rates,
lacks evidence of Quaternary tectonic uplift. Rather, the reported recent tectonic activity is accommodated
by strike-slip movements with no significant relief building (e.g., Ait Brahim et al., 2002; El Azzab and El
Wartiti, 1998; Fedan and Thomas, 1986; Hinaje et al., 2019). As a matter of fact, unlike the eastern flank of
the Middle Atlas, the western one is devoid of a well-developed synorogenic flexural foreland basin,
suggesting that tectonic shortening is primarily accommodated by thrusting along the SMAF. This could be
explained by late Neogene partitioning of oblique deformation, such that strike-slip kinematics was
accommodated along the NMAF, while thrusting occurred along the SMAF (Gomez et al., 1998).
Furthermore, our field observations suggest that the Quaternary lava flows cover the major faults in
western side of the Middle Atlas, i.e. Tizi N'Tghatten Fault (Fig. 9A) and the North Middle Atlas Fault (Fig.
7B). This indicates that the low post-lava emplacement incision rates calculated on the western flank of the
Middle Atlas could reflect the lack of substantial Quaternary thrust-related surface uplift.

Additionally, the Quaternary incision rates in the Middle Atlas are also lower than those attributed to

the mantle dynamic uplift estimated by Pastor et al. (2015). Interestingly, despite the large wavelength of
this component of surface uplift (Clementucci et al., 2023b; Frizon de Lamotte et al., 2009; Lanari et al.,



2023; Missenard et al., 2006; Pastor et al., 2015), the Quaternary incision rates in the Middle Atlas do not
appear to record such an uplift. This disparity could be explained as a fluvial response to the mantle-related
uplift documented in the Massif Central (Clementucci et al., 2023b). It appears that this perturbation (e.g.,
erosional wave) has not yet reached the localities where we measured post-lava emplacement incision
rates, resulting in the topography remaining in a transient state. This is consistent with the fact that most of
our lower incision rates were calculated upstream of the non-lithological knickpoints (Figs. 7, 8, 9 and 10).
In contrast, the incision rates calculated in the Oum Rbia River represent an exception as they are of the
same order of magnitude as elsewhere in the Middle Atlas although being measured downstream of the
river major knickpoint (Fig. 7). This knickpoint may not be linked to mantle-related uplift but to a base-level
fall due to drainage reorganization of the Oum Rbia River during the late Pliocene (Yaaqoub and Essaifi

2023).

Another possibility for explaining the slower post-lava emplacement incision rates in the Middle Atlas

with respect to the Moroccan Massif Central could be that localized surface uplift is affecting exclusively
the Massif Central. Clementucci et al. (2023b) identified a phase of surface uplift in the Massif Central
preceding the regional deep-mantle-related uplift (Babault et al., 2008). This phase may have started in the
early Miocene, with ~400 m of surface uplift that is interpreted to reflect crustal-scale processes (e.g.,
magma injection or mantle delamination; Clementucci et al., 2023b). An alternative possibility is
considering a flexural component contributing the total uplift. Given the regional tectonic context of the
Massif Central, which is situated to the south of the flexural Rharb and Saiss foredeep basins (Fig. 14;
Chalouan et al., 2008; Flinch, 1994), a forebulge flexural uplift related to loading in the Rif Belt may have
been a contributing driving mechanism for this phase of surface uplift in the Massif Central. Forebulge
uplift due to the tectonic overburden of Rif Cordillera is also recognized in the southern Rharb shelf (Le Roy
et al., 2014) and supported by independent evidence such as the flexural extension in the Rharb basin
(Zouhri et al., 2002) and the recent extensional deformation in the Rharb-Saiss basins coeval with
compression in the Western Meseta (Bargach et al., 2004). Furthermore, flexural uplift in the Massif Central
likely began during the middle-late Miocene, given the initiation of subsidence in the foredeep during this



time (Chalouan et al., 2008). Forebulge uplift has been demonstrated as one of the mechanisms driving
topographic rejuvenation across various forelands worldwide (*e.g.,* Expósito et al., 2022; Nivière et al.,
2013; Repasch et al., 2023). Typically, the amplitude of forebulge uplift is 4% to 7% of foredeep maxima
(Decelles, 2012). Considering the mean of these percentages (5.5%) and a maximum of 2000m for thickness
of the Saiss foredeep (Sani et al., 2007), the amplitude of the forebulge uplift in the Massif Central would be
110 m. This value is consistent with the incision heights measured in the Masssif Central (Table 1) and near
to the predicted forebulge uplift (174 m) in the Sierra Morena at the foreland of the Betic Cordilleras
(Garcia-Castellanos et al., 2002), which are the northern limp of Gibraltar Arc. However this estimated
forebulge uplift (110 m) cannot solely account for the high elevation of Massif Central and is inconsistent
with the values of surface uplift (400 m) estimated from river profiles projections for the same area
(Clementucci et al., 2023b). Therefore, we suggest that the lithospheric flexure may have facilitated inflow
of asthenospheric material from the adjacent Middle Atlas towards the Massif Central during the
Quaternary (Fig. 14), thereby contributing to localized dynamic topography (*e.g.* Miller and Becker, 2014).
This hypothesis finds support in the fact that the alkaline volcanism, associated with mantle dynamics,
started in the Middle Atlas before beginning in the Massif Central (El Azzouzi et al., 2010; Rachdi, 1995).
Moreover, a late Pliocene loss in the drainage area of the Bou Regreg River decreased erosion efficiency in
the Massif Central, preventing it from keeping pace with the sustained uplift. This likely fostered localized
surface uplift in the Massif Central (Yaaqoub and Essaifi, 2023). Therefore, the relatively high, post-lava
emplacement incision rates in the Massif Central could be the result of an incision wave that migrated
upstream from the Atlantic Ocean in response to flexural surface uplift.

### 5.4. Quaternary Landscape evolution
The Quaternary fluvial incision rates of the Middle Atlas and the Massif Central allow us to
constrain the amount of relief generated in this region during the Quaternary and to gain insights into the
landscape evolution before and after the lava flows emplacement. River incision rates in lava-filled valleys
in the Middle Atlas range between 8 and 38 m/Ma, (i.e., only a few tens of meters in a million year),



whereas up to one hundred meters of fluvial incision for the same time frame is calculated for the Massif
Central. This indicates that the post-lava emplacement incision in the Middle Atlas has been a relatively
slow process during the Quaternary. This inference is reflected in the regional landscape when we consider
the amount of relief generated after lava emplacement compared to the total relief of lava-filled valleys.
Overall, the post-eruption incision represents only a small proportion of the total fluvial incision (Fig. 12, 13
A and C). Consequently, the calculated post-lava emplacement incision rates indicate that the greatest part
of topographic relief in the Middle Atlas predates the Quaternary. For instance, lacustrine deposits
attributed to the Mio-Pliocene (Beaudet, 1969) have undergone substantial erosion that led to their
preservation in the form of buttes dispersed throughout the western boundary of the Middle Atlas where
their heights reach hundreds of meters. Importantly, this significant erosion occurred before the
emplacement of Quaternary lava flows (Fig. 12). Similar buttes topped by lacustrine and continental
sediments shaped before the emplacement of Quaternary lavas are also observed in headwaters of the Bou
Regreg catchment (Yaaqoub and Essaifi, 2023). Furthermore, a petrified lava cascade has been described at
the transition by Martin (1981) between the TMA and the Western Meseta (Fig. 12 C), indicating the
control of preexisting relief on the Quaternary lava flows emplacement.













**Figure 12A) and B) Cross-valley topographic profiles highlighting the total and Post lava emplacement topographic relief in the Middle Atlas and Western Meseta respectively. C) Perspective view showing the control of preexisting relief on lava flow emplacement.**




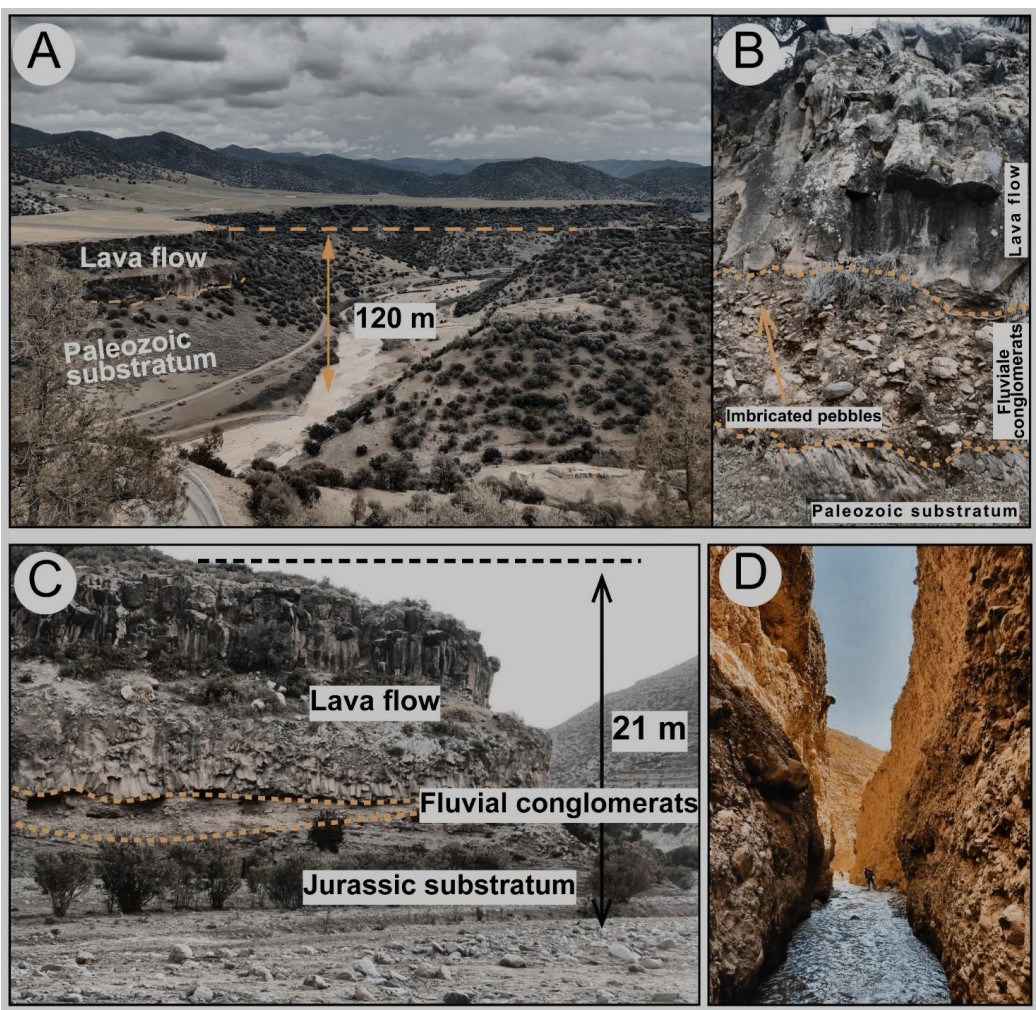

**Figure 13: Field Photographs, A) landscape around Bou Tsaggatine locality, B) Left valley side at the Bled Belmadi locality showing a succession of Paleozoic basement, fluvial conglomerates, and lava flow, C) Right valley side at Taghazout locality with Jurassic substratum, fluvial conglomerates and lava flow, D) Gorges of Cheg El Ard River in the eastern flank of the Middle Atlas.**




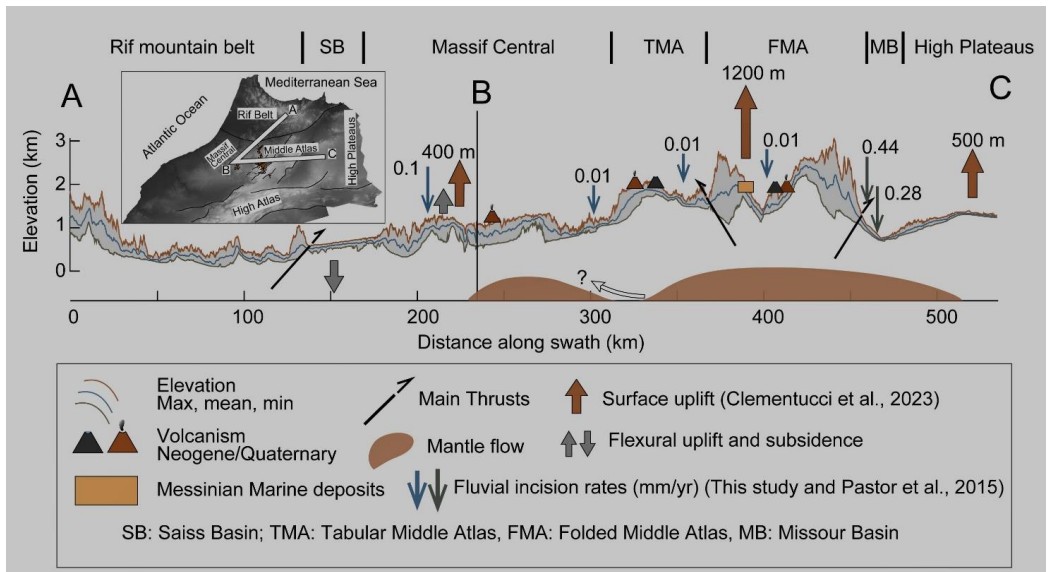

Figure 14: Schematic topographic swath profile summarizing the main data on surface uplift and incision rates across the study area. The inset shows the swath location.


## 6. Conclusion


The Quaternary fluvial incision rates in the Middle Atlas and the Massif Central inferred from lava
flows, combined with geomorphic analysis, provide insights into the regional evolution of the geomorphic
processes and the uplift history. The observed spatial discrepancy in incision rates, with values of
approximately 0.01 in the Middle Atlas and 0.1 in the Massif Central, is consistent with variations in
geomorphic metrics such as local relief, $Ksn$, and hypsometry. These metrics indicate higher geomorphic
complexity in the Massif Central as compared to the Middle Atlas. Furthermore, a comparison of our
incision rates with previously published fluvial incision rates in the northeastern flank of the Middle Atlas,
suggests a dichotomy in the tectonic evolution of the Middle Atlas range with its eastern flank that is
primarily accommodating active tectonic processes.



The calculated fluvial incision rates do not appear to be influenced by lithology or climate. Instead,
the spatial distribution of surface uplift inferred from river projection in previous studies seems to strongly
influence the spatial variations in fluvial incision rates. Specifically higher Quaternary incision rates are
systematically downstream of major non-lithological knickpoints indicating that the elevated upstream
portions of the landscape have not yet adjusted to an acceleration in surface uplift rates. High fluvial
incision rates in the Moroccan Massif Central can be related to surface uplift due to the flexural uplift as a
forebulge in the foreland of the Rif Belt, enhanced by the deep mantle-related uplift. These findings
contribute to our understanding of the geomorphic and tectonic dynamics in the Middle Atlas and Western
Meseta regions during the Quaternary and highlight the importance of lava flows as geomorphic markers.
## Author contributions
AY and AE Conceptualized the study and collected the data. AY analysed the data and wrote the original
draft. AE, RC, PB, RZ, CF and CP discussed the findings, reviewed and edited the manuscript.
## Competing interests
The authors declare that they have no conflict of interest.
## Acknowledgements
This work is part of a PhD program. The PhD student, AY, acknowledges the Moroccan National Center for
Scientific and Technical Research (CNRST) for awarding him a scholarship (21UCA2019). (…)

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
