# Peer review of "Geomorphic analysis and fluvial incision rates from valley-filling lava flows: implications for the Quaternary morphotectonic evolution in the Moroccan Massif 2 3 Central and Middle Atlas Ahmed Yaaqoub¹, Abderrahim Essaifi¹, Romano Clementucci².³, Paolo Ballato², Rachid Zayane¹, Claudio 4 5 Facc"

_EGUsphere, 2024_

## Author Comment (AC1)

**Ahmed Yaaqoub,**

Cadi Ayyad University, Marrakech, Morocco.

To: Dr. Prof. **Veerle Vanacker**,

Handling Editor, Earth Surface Dynamics

**Object:** Response to referees' comments and suggestions made regarding the submission " *Geomorphic analysis and fluvial incision rates from valley-filling lava flows: implications for the Quaternary morphotectonic evolution in the Moroccan Massif Central and Middle Atlas*" [EGUSPHERE-2024-1746] by A. Yaaqoub, A. Essaifi, R. Clementucci, P. Ballato, R. Zayane, C.Faccenna, C. Pagli.

Dear Dr. Prof. **Veerle Vanacker**,

First, we sincerely thank the reviewers and yourself for your time and effort in providing insightful and constructive feedback. Below, we address each comment point by point, with our responses highlighted in red text. We will also upload the revised manuscript and hope that the improvements align with the standards of Earth Surface Dynamics.

Best regards,
Ahmed yaaqoub, on behalf of all co-authors.

**Reponses to referees' comments:**

**Reviewer #1: Anonymous:**

The study by Yaaqoub et al. presents a geomorphic analysis of the Moroccan Massif, and Central and Middle Atlas. This is combined with post-lava emplacement incision rates to study the drivers of landscape evolution. The incision rates presented are interesting, especially when compared to the previously published, faster estimates of regional uplift. However, the study lacks a clear research question or hypothesis and the writing is often goes into excessive detail, making it read more like a textbook than a research article. The analysis is similarly unfocused, with a lot of plots and text, but limited information being conveyed. Despite these issues, the incision data appear robust, and there are several noteworthy observations that, if properly emphasized, could significantly strengthen the paper. Below, I provide a list of suggested improvements (in no

particular order), along with detailed line-by-line comments that I hope the authors will find constructive for future versions of this manuscript.

Thank you so much for your relevant remarks, we hope these revisions below will address your concerns and strengthen the manuscript. The line numbers in the replies are from to the revised manuscript.

The background section includes a large amount of unnecessary information, which is more distracting than useful. Also, because the reader is not informed about the research question/hypothesis/approach of the study in the introduction, it is hard to judge what information is relevant while reading. Please, only include information relevant to the study at hand.

Thank you, we modified the revised text accordingly. Now, the geological setting section includes only the most relevant information. Moreover, the introduction states clearly the research question of this study. Please see lines 74-83.

The methods section also includes A LOT of unnecessary detail. This level of detail is not required for an ESurf submission. The authors should focus on the important details, such as the parameters used, etc. Some examples are found in lines 183-184, 187-194, 202-208.

Thank you! We have in the revised methods section we provide only the essential methodological parameters. I hope this has improved the clarity and conciseness of the text.

In the case where lava flows pour into valleys of weaker bedrock, stream incision may try to avoid the hard volcanic rocks and focus on the contact between non-volcanic bedrock and the lava flow. Does this occur here too? Nothing like this was mentioned in the text, but I'd assume that the sedimentary section of the plateau is easier to erode than the lava flow.

Thank you for your insightful remark.

The location of incision should not influence the icision rates once the river will have a reached an equilibrated profile. Indeed incision may start along the contact between 2 lithologies because that is a weakness. What should be expected, however, is that at the lava front a knickpoint (knp) will start propagating upstream along the lava flow until a new equilibrated longitudinal profile will be reached. River segments downstream of this knp will have an incision rate set by the regional/local background uplift. This rate will be independent from the location of fluvial incision.

A softer lithology will only impact the speed at which the knp travels upstream $dz/dt=K*A^m$ where where, $dx/dt$ is the knickpoint celerity, $K$ is a dimensional coefficient of erosion $A$ is upstream drainage area and $m$ is a non-dimensional parameter that depends on basin hydrology, channel geometry, and erosion process (Whipple, & Tucker, 1999). That said, we consistently avoided measuring incision at these contacts lava-bedrock. However, there is one exception: at site N°06, the river incises along the contact between the Quaternary lava flow and Eocene limestone (see figure below). Despite this, the incision rate measured at this location (0.02 mm/year) is consistent with the low rates observed elsewhere in the Middle Atlas and remains significantly lower (by ten) than those in the Central Massif.

[Figure]

This observation supports our conclusion that the spatial variation in incision rates is not primarily controlled by localized erosion at lava-bedrock contacts but rather by broader regional factors.

Throughout the paper, especially in the result section, a lot of reference is made to local town names. As someone unfamiliar with the region that makes the text harder to follow and provides an unnecessary amount of detail. It would be better to explain the large-scale patterns using the names of just a few domains: Folded Middle Atlas, Tabular Middle Atlas, etc., Lines 280-193 are a good example, where the paragraph starts fine and then dives into a lot of detail, without a takehome message.

Thank you for your remarks, we have tried to minimize as much as possible the reference to local names.

The results section goes through several river catchments in a lot of detail, but it remains unclear what knowledge is gained from the individual analysis. The paper would be much stronger if all the geomorphic analysis was condensed to 2-3 figures and paragraphs that lay out the general patterns, which then can be compared to the incision rate patterns.

Thank your for your suggestion. We decided to keep most of the figures and the catchment analysis because we think that this structure is necessary for a correct understanding of our

study. Nevertheless, to increase the readability of the manuscript we moved Table 1 and former Figure 3 (the stratigraphic column) to the Supplementary materials section.

The incision rates are the most interesting part of the paper. However, they are currently underused, while the geomorphic metrics are overused. For instance, there is no map showing the spatial distribution of incision rates. Incision rates from lava flows could be compared to the Ksn map to evaluate if the topographic pattern matches the actual incision rates.

Thank you for this critical observation. In the revised text we plotted the location of the sites used for estimating lava flow incision rates on the $K_{sn}$ map of figure 5. However there, we did not report the incision rates because they would have created a vey dense and unreadable figure. Instead, the rates are reported for each catchment in map view and along the longitudinal river profiles in figs. 6 to 9, as well as in map view in the summarizing figure 10. In figure 6 to 9 the spatial distribution of the rates with respect to the channel geometry (both the spatial and in the "chi" space) can be appreciated.

All figures are underlain by a dark shading, which makes it harder to see things on the maps.

Thank you for pointing this out. The dark shading is not removed in the revised figures.

Line 17-19: The wording in the second sentence should be revised. Currently, it is unclear what noun is referred to when the authors write: are primarily controlled by climate. I assume they refer to the fluvial terraces but this is unclear.

Thank you for pointing that out. The last phrase referred to 'surface processes,' but we have removed it for clarity (line 16).

Line 30: How does this comparison point to active shortening? No reason is stated.

This sentence is reformulated for more clarity (lines 26-28).

Line 33-37: No data or arguments are presented to support the arguments. As a reader, I am left in the dark.

Thank you for your remark. We have revised the text to clarify our argument (lines 31-39). We now explicitly state that the high elevation of the Massif Central, despite the absence of Neogene shortening, suggests uplift driven by forebulge uplift enhanced by asthenospheric upwelling. We also clarify how this uplift contributed to river incision by lowering base levels and specify that our conclusion on topographic relief development is based on a comparison of pre- and post-lava landscapes.

Line 67: This sentence is very confusing. The late Miocene and Pliocene are part of the Neogene. Therefore, I am confused what "in the Neogene before the late Miocene and during the Pliocene"

means. Were there two phases of mountain building, one in the early-middle Miocene, and onein the Pliocene?

Sorry for the confusion. The sentence was modified. We mean shortening occurred during two phases: one in the early-middle Miocene and one during the Pliocene (lines 66-67).

Line 89-90: Check language. Maybe "and is part of the Cenozoic Alpine belt".

Changed thank you (line 100).

Line 94: What does alpine-type refer to?

It refers to the Moroccan Rif interplate mountain belt. At the opposite of the Atlas intraplate mountains. In the revised text we removed '-type'.

Line 152: Typo. Messinian.

Corrected, thank you.

Line 152-155: This sentence is written as if there was a contradiction between this and the previous statement. It seems like the authors meant to say uplift may have started pre-Messinian. But the sentence says "in the late Cenozoic", which would include the Messinian.

Sorry for the confusion. This sentence is removed.

Line 219: Why modelling? It seems like the actual profiles are being analyzed and no modelling takes place. Please, rephrase.

This phrase was deleted.

Line 220-223: This is only true for detachment-limited bedrock rivers. Please, also add a statement that addresses whether detachment-limitation is a reasonable assumption in your study area.

The detachment-limited condition is likely valid for rivers draining bedrock and where the stream power and transport capacity is higher than sediment load supplied from upstream. This roughly defines the capacity of river to incise and/or depositing. However, in this study we only considered rivers that are incising to some degree the landscape. We add a sentence and rephrased this concept in the method section 3.2 and 3.3.

Line 224: What is "the later model". Do you mean "latter"? If so, please add that the equation that is being shown only emerges if the stream power incision model is combined with a conservation of mass statement.

"The later" was deleted. The sentence was corrected according to the comment of the reviewer (line 222).

Line 231: Only if you solve for slope.

Right, thank you, we emphasized that (lines 229-230).

Line 234: Typo. Empirical.

Corrected, Thank you

Line 235: The s of $k_s$ is not in the subscript.

Corrected, Thank you.

Line 254: change capital K in Ksn to lower case k.

Changed, thank you.

Line 238: Deviations from what form? Do the authors mean when ks changes along profile? That can also be linked to numerous other factors, e.g., a change in K.

Sorry for the confusion. We meant the deviation of the river profile from the concave-up shape. We have improved the clarity of this sentence. See lines 235-237.

Line 258: Which knickpoints? So far there was no mentioning of knickpoint mapping. No indication has been given of how they were mapped, and what criteria were used to map them.

Thank you for your comment. We have clarified that the identification of knickpoints is based on slope changes in Ks within the longitudinal profiles and χ plots (see lines 253-254).

Line 260 and following: references are made to colors, but here is no reference to the relevant figure.
Thank you for pointing this out. The references to the relevant figures were added.

Line 275: What dating methods were used for these lava flows?

The K/Ar is the method used for dating. This information is added in the line 270.

Line 473-476: It depends whether the landscape it thought to be near steady state or not. Here, one could use the river profile to argue for transience, as well as pointing out that the velocity of knickpoints is dependent on river discharge. Hence, climate may play a role.

We thank the reviewer for the comments. The sentence has been modified accordingly. Moreover, another aspect that has been introduced in the revised text is erodibility (lines 460-467).

Line 504-505: The period of observation stated in the figure caption is 1901-1981 and therefore different than what is said here.
It was an error, thank you for pointing this out. The period is 1981-2011.

Section 5.2: It would be more informative to plot incision rate against mean annual precipitation instead of a qualitative map interpretation. Also, what about climate change? Climate is only addressed in terms of spatial patterns, but what about temporal changes? Later in the paper the difference between incision rates from lava's (shorter time-scale) and from uplifted sediments (longer time scale) is discussed. Could the slower lava incision rates be related to increasing aridity?

Thank you very much. We have created a new figure where we plot rainfall against incision rates and integrate it with rainfall map (Fig. 11B). Unfortunately, we lack data on past climate changes in study area. However, the literature indicates that the Quaternary climate was characterized by alternating pluvial and interpluvial periods, corresponding to phases of abundant precipitation and drier conditions, respectively (Awad, 1963; Wassenburg et al., 2012). These climatic cycles are believed to have affected all of Morocco (Awad, 1963). Therefore, we consider them unlikely to explain the spatial variations in post-lava incision rates that we observed.

Section 5.3 It is extremely hard to follow these dense paragraphs loaded with specific place names. A lot of the information presented here can be streamlined.
Thank you for your remark; we split it into several paragraphs and tried to enhance it clarity.

Line 620: This is an interesting hypothesis and should be explained in more detail. Maybe also a schematic.
This hypothesis is discarted. The idea is that flexural uplift is enhanced by deep-mantle uplift as illustrated in figure 14, which we slightly improve to more readability.

Line 625: Wait. How does the drainage capture no come into play? Why is the drainage capture as mechanism mentioned and then immediately neglected in the concluding statement of the paragraph?
Here, we mention that the drainage area loss of the Bou regreg River due to a river capture that took place during the Pliocene, would have favored the enhancement of the already sustained surface uplift. We did go into detail because this scenario is explained in one of our previous paper (Yaaqoub and Essaifi , 2023) to which we referred for me details. That said, that part is removed from the revised manuscript.

Line 640: Doesn't this support the hypothesis of aridification leading to a decrease in incision rates? Could this be an interesting research question to ask?

As mentioned earlier, we lack sufficient paleoclimatic data for our study area, which would be necessary to determine whether significant climate changes have influenced incision rates. Therefore, the hypothesis of aridification affecting incision rates cannot be verified at this time.

Fig. 1 Area of thinned lithosphere and drainage divides are hard to see. Either use lighter colors for these labels or a lighter color map to increase contrast. My suggestion would be to replace the color map with one that is perceptually uniform and color blind friendly.

Thank you for your suggestion to enhance the visibility of the figure. We changed the color of the digital elevation model to a lighter one.

Fig. 2: Studied valleys with lava flows and dating locations should be added to this figure.

Thank you for your suggestion. We have added the studied valleys with lava flows and dating locations to the figure.

Figures 12 and 13 are great and I wish they would have come much earlier in the manuscript!

Thank you! We have moved Figures 12 and 13 to Section 5.4 so they now appear earlier in the revised manuscript.

Please, describe what can be seen in figure 13D.
Done, thanks.

Figure 14: Can you split this into 2 cross-sections? The change in direction of the profile makes it harder to understand.

Thank you for your suggestion. To improve the clarity, we have slightly rotated the first profile to give a more perspective-like view. We hope this adjustment makes the figure easier to understand.

Line 672: What is geomorphic complexity?
This is changed to 'greater landscape dissection'.
* * *
**Reviewer #2: Jo De Waele**

Dear authors,

I read your ms with interest, and while doing so made many corrections on the pdf file (attached). Most are minor things, that will make you help "shaving" the ms a bit.

I know the areas you studied, so I more or less got through the ms comfortably, but I am sure several readers will get lost in the jungle of names and descriptions. Not sure how to deal with this. I agree with the other reviewer on many points he (or she) rises, and will not come back to these...I am also not an expert in fluvial geomorphology...

The methods you use appear to be adequate, and it all is based on an extensive literature review. I have some major concerns, which I list below.

Thank you for your time and effort in providing these valuable remarks. We appreciate your feedback and hope that the revisions below effectively address your concerns and enhance the manuscript. The line numbers in our responses refer to the revised manuscript.

1. my major one: you never take into account the karstic nature of parts of the study area. In karst areas, water mainly follows underground routes, and the erosional potential is focussed along subterranean flow routes, rather than at the surface in streams. This is completely ignored! The Middle Atlas is entirely karstic, so of course erosion rates in surface rivers is lower than in the siliciclastic bedrocks. This undermines a big part of your discussion and conclusions: what you see is really (entirely) due to differential uplift between areas? Are lithology and climate completely uninfluential on fluvial incision rates? I would need to be clearly convinced this is the case...

Please: acknowledge that karst needs to be considered; describe the areas for their karst; maybe put some data on present surface runoff (flowrates) in the rivers you have investigated (explain hydrology of aquifers and rivers more in detail); put this also in a geochronological and climatic times scale and perspective (rainfall and runoff has not been constant during the Quaternary, influencing also karst!). Then convince me that the fact certain areas are karstic has no influence on fluvial erosion rates...!!

Thank you very much for raising this important point regarding the karstic nature of parts of our study area. Your observation is well-founded, as water in karstic settings predominantly flows underground, which would naturally reduce the efficiency of surface fluvial incision.

At first glance, there appears to be a correlation between karst and incision rates as: the Central Massif, where we measured higher post-lava incision rates, is composed of non-karstifiable Paleozoic metasediments, while the Middle Atlas, where incision rates are lower, is

predominantly made of carbonates prone to karstification. Hydrological studies also confirm that the aquifers in the Middle Atlas are karstic (e.g., Akdim, 2015). However, this apparent correlation is not consistent across the entire study area. Specifically:

1. Beht River case:

Two of our post-lava incision rates (sites N° 9 and 10, Fig. 8) were measured in the Beht River, which drains the northern part of the Central Massif. Although the substrate here is non-karstifiable and similar to the Bou Regreg catchment (where the highest incision rates were calculated), these two locations exhibit low incision rates. This suggests that karstification alone cannot explain the observed spatial variability in incision rates.

2. Oum Rbia River case:

The post-lava incision rates along the Oum Rbia River are consistently low compared to those of the Bou Regreg River, regardless of whether they are measured in the upstream karstified carbonate areas or the downstream non-karstifiable Triassic and Paleozoic terrains. Notably, even though incision rates (sites N° 1, 2, 3, and 4, Fig. 6) were calculated downstream of the Oum Rbia sources—which are among the most important in Morocco in terms of discharge—they still exhibit low values. This further supports the idea that karstification is not the primary factor controlling incision rates.

Based on these counterexamples, we suggest that karstification cannot be the primary factor explaining the spatial discrepancy in incision rates across the study area. However, we do not discount its potential influence locally, particularly in the Middle Atlas. We agree that a focused study would be valuable to better understand the impact of karstification on surface fluvial incision in this region.

That said, in response to your comments, in the revised text we: i) Described the karstic nature of the Middle Atlas in the geological setting section of the manuscript (lines 144-152)and ii) Discussed the potential influence of karstification on post-lava incision rates in the lithological subsection (Subsecion 5.1)(Lines 482-498).

2. My second comments regard the SRTM data and their resolution (altitudes). Are they precise enough to draw reliable conclusions? Can you better describe in methods how useful these data can be in your analysis?

Thank you so much for your remark. The SRTM 90 m DEM used in this study has a vertical resolution of ±16 m at a at 90% confidence level (Farr et al., 2007). While this level of precision may not be suitable for applications requiring extremely high accuracy (e.g., engineering studies), we think it is precise enough for regional-scale geomorphic analyses such as ours, which focus on

broad topographic patterns and fluvial processes across a large area (Middle Atlas mountain belt and Central Massif Plateau). In our study, the SRTM 90 m DEM was used to generate topographic metrics such as local relief, swath profiles, and river longitudinal profiles. These metrics were combined with post-lava emplacement fluvial incision rates estimated on the field, which are the primary focus of our analysis. Importantly, the incision rates were not derived from the DEM but were independently measured in the field, ensuring that the vertical resolution of the SRTM data did not directly affect these critical measurements. Finally, our field-derived incision rates and DEM-derived topographic metrics are consistent, which further supports the reliability of our results.

3. Scenarios: your conclusions are very focussed, with no description of different possible explanations: I ask you to describe the different possible scenarios, describing pros and cons of each, and then "promote" the most probable one. Just an example (but there could be many more): the Plio-Quaternary volcanic activity has brought lavas to the surface, creating voids underneath that might have caused the same areas to subside... is this possible? Why or why not? Has it been considered? During ice ages, the Middle Atlas was covered with glaciers: has this caused glacio-isostatic rebound? Many other possible vertical movements would need to be discussed....

Thank you for your suggestions—they are certainly interesting. However, we have limited our discussion to the three main factors/scenarios that are most relevant and frequently invoked in tectonic geomorphology studies.

The scenarios you proposed are indeed intriguing and could have influenced incision. However, we do not believe they are significant in the context of a regional study such as ours. For instance, the volume of volcanic material produced by Plio-Quaternary activity in the study area is relatively small. In the Middle Atlas, for example, volcanic deposits are estimated to have a total volume of approximately 20 km³, distributed over a wide area of about 1,000 km² (El Azzouzi et al., 2010). Given this limited volume, we do not believe it could have caused substantial subsidence-related vertical surface movements and at least there are any strong evidences of local subsidence or uplift of the bottom surface where the volcanic centers are located.

Glacio-isostasy is also a fascinating topic that warrants thorough investigation. Unfortunately, there is a scarcity of data regarding Quaternary glaciation in the study area. The only documented glacial and periglacial features are observed in the northeastern part of the Folded Middle Atlas (Awad, 1963; Beaudet, 1971; Dresch and Raynal, 1953; Hughes et al., 2011; Raynal, 1953), which is not covered by our post-lava incision rate analysis. Even if we assume a component of surface uplift related to glacial isostasy, it would have led to increased incision rates—an effect that is not supported by our data, as post-lava incision rates remain low. Given the lack of robust data on the

extent and significance of Quaternary glaciation in the Middle Atlas, any discussion of this factor would be purely speculative.

**References:**

Akdim, B.: Karst landscape and hydrology in Morocco: research trends and perspectives, Environ. Earth Sci., 74, 251–265, https://doi.org/10.1007/s12665-015-4254-5, 2015.

Awad, H.: Some Aspects of the Geomorphology of Morocco Related to the Quaternary Climate, Geogr. J., 129, 129, https://doi.org/10.2307/1792631, 1963.

El Azzouzi, M., Maury, R. C., Bellon, H., Youbi, N., Cotten, J., and Kharbouch, F.: Petrology and K-Ar chronology of the Neogene-Quaternary Middle Atlas basaltic province, Morocco, Bull. la Société Géologique Fr., 181, 243–257, https://doi.org/10.2113/gssgfbull.181.3.243, 2010.

Beaudet, G.: Le Quaternaire marocain: état des études, Rev. Géographie du Maroc, 1971.

Dresch, J. and Raynal, R.: Formes glaciaires et periglaciaires dans le Moyen Atlas, Verlag nicht ermittelbar, 1953.

Farr, T. G., Rosen, P. A., Caro, E., Crippen, R., Duren, R., Hensley, S., Kobrick, M., Paller, M., Rodriguez, E., Roth, L., Seal, D., Shaffer, S., Shimada, J., Umland, J., Werner, M., Oskin, M., Burbank, D., and Alsdorf, D.: The Shuttle Radar Topography Mission, Rev. Geophys., 45, 65–77, https://doi.org/10.1029/2005RG000183, 2007.

Hughes, P. D., Fenton, C. R., and Gibbard, P. L.: Quaternary Glaciations of the Atlas Mountains, North Africa, in: Developments in Quaternary Science, vol. 15, Elsevier Inc., 1065–1074, https://doi.org/10.1016/B978-0-444-53447-7.00076-3, 2011.

Raynal, R.: Deux exemples régionaux de glaciation quaternaire au Maroc, Haut Atlas Orient. Moyen Atlas Septentr. Actes IV, Congrès Int. du Quat. Rome-Pise, 1953.

Wassenburg, J. A., Immenhauser, A., Richter, D. K., Jochum, K. P., Fietzke, J., Deininger, M., Goos, M., Scholz, D., and Sabaoui, A.: Climate and cave control on Pleistocene/Holocene calcite-to-aragonite transitions in speleothems from Morocco: Elemental and isotopic evidence, Geochim. Cosmochim. Acta, 92, 23–47, https://doi.org/10.1016/j.gca.2012.06.002, 2012.

Whipple, K. X., & Tucker, G. E.: Dynamics of the stream-power river incision model: Implications for height limits of mountain ranges, landscape response timescales, and research needs., J. Geophys. Res. Solid Earth, 104(B8), 17661–17674, 104, 661–674, https://doi.org/https://doi.org/10.1029/1999JB900120, 1999.